

# Iterative feedback tuning of wind turbine controllers

Edwin van Solingen[1] and Jan-Willem van Wingerden[1]

[1]Delft Center for Systems and Control, Faculty of Mechanical Engineering, Delft University of Technology, Mekelweg 2, 2628 CD Delft, The Netherlands

*Correspondence to:* Jan-Willem van Wingerden (J.W.vanWingerden@tudelft.nl)

**Abstract.** Traditionally, wind turbine controllers are designed using first-principles, linearized, or identified models. The aim of this paper is to show that with an automated and model-free tuning strategy, wind turbine control performance can be significantly increased. To this purpose, Iterative Feedback Tuning (IFT) is applied to two different turbine controllers: drivetrain damping and collective pitch control. The results, obtained by high-fidelity simulations using the NREL 5MW wind turbine, indicate significant performance improvements over baseline controllers which were designed using classical loop shaping techniques. It is concluded that iterative feedback tuning of turbine controllers can potentially become a valuable tool to improve wind turbine performance.

## 1 Introduction

The control system plays a crucial role in the operation of wind turbines (van Kuik et al., 2016). Without properly tuned control loops, the turbine does not extract the maximum amount of energy from the wind and loads are not mitigated. Typically, wind turbine controllers are designed using linearized models obtained from wind turbine software packages (Bossanyi and Witcher, 2009). The linearized models approximate the nonlinear wind turbine dynamics in the vicinity of selected operating points for which the controller is designed. To obtain a controller which performs satisfactory across the different operating conditions often gain-scheduling techniques are used. When necessary, the controllers can be fine-tuned by connecting them to the nonlinear wind turbine model using high-fidelity software packages.

Several factors detriment the controller performance when implemented on the actual turbine. First, the controller is designed upon the basis of models. This directly implies that there will be modeling errors and, hence, differences with the actual turbine. Second, every turbine will be different from the specifications due to for instance manufacturing errors and imperfections (van der Veen et al., 2013a). Third, due to environmental differences such as varying soil dynamics throughout a wind plant, the dynamics of wind turbines vary per turbine (Lombardi et al., 2013; Abhinav and Saha, 2015). Finally, due to wear and tear, dynamics will change over time. All these factors impact the (controller) performance of wind turbines and should be addressed during commissioning and periodically during the lifetime of a wind turbine.

A wind turbine manufacturer has several opportunities to overcome the aforementioned issues. One of these is by applying system identification techniques (Hjalmarsson, 2005; van der Veen et al., 2013a, b; La Cava et al., 2016). With system identification, the dynamics of the actual turbine are obtained by exciting the wind turbine and measuring the response thereof, which often yields a more realistic wind turbine model. By using the identified models as a basis for the control design, perfor-



mance can be increased. Drawback of this approach is that it might be time consuming to obtain the dynamics for all operating conditions, after which the controller needs to be redesigned.

In the past decade, research has been conducted on developing load reducing controllers that 'learn' the optimal controller settings online (van Wingerden et al., 2011; Houtzager et al., 2013; Navalkar et al., 2014, 2015; Xiao et al., 2016). These con-
trollers are typically scheduled on basis functions and thereby mainly target the periodic wind turbine loading. By minimizing a cost function with respect to the controller parameters, the optimal (linear) combination of the basis functions can be obtained online. The controllers have been successfully demonstrated both in simulation studies as well as in experimental wind tunnel testing. However, the main drawback of the majority of the approaches is that the resulting controller operates in feedforward. This means that mainly the deterministic loads are targeted, while stochastic loads remain roughly unaffected.

Another strategy is to tune the controllers offline by using the previous mentioned models, and use this as starting point for an automated online tuning algorithm (which can for instance be ran during commissioning and periodically over the turbine lifetime). One such algorithm is given by Iterative Feedback Tuning (IFT) (Hjalmarsson et al., 1998; Hjalmarsson, 2002). With IFT, the controller parameters of a controlled system are iteratively optimized by carrying out two or more experiments on the closed-loop system, with which estimates of the gradient are obtained. Iteratively updating the controller parameters
by using the gradient estimates then minimizes a user-defined cost function. The key advantage of IFT is that detailed knowledge of the (wind turbine) system is not needed, while the main requirement is that the initial closed-loop system is stable, implying that it can be directly used for optimization of wind turbine controllers. IFT has been successfully applied to various application fields including mechatronics (Al Mamun et al., 2007; Heertjes et al., 2016), robotics (Liu et al., 2011), process industry (Lequin, 1997), and recently also wind turbines (Navalkar and van Wingerden, 2015).

In this paper, we use IFT to optimize the performance of an active drivetrain damper and of reference tracking using Collective Pitch Control (CPC). The main contribution is therefore to show the (practical) application of IFT to existing wind turbines. The paper also contributes in showing how IFT can be applied to systems that have multiple controllers in the loop. Moreover, it is shown that IFT can be applied to systems that have a reference input with a static offset, by performing an additional experiment. Finally, the impact of several practical considerations are shown including the experiment length, signal-to-noise
ratio, and convergence speed.

The paper is organized as follows. In the next section, the details of the IFT algorithm, and the closed-loop experiments that are required to obtain gradient estimates, are given. Subsequently, in Section 3, short descriptions of the NREL 5MW wind turbine and the high-fidelity simulation environment are given. An overview of the control system is also provided. In Section 4 and 5, the IFT algorithm introduced in Section 2, is used to optimize the performance of an active drivetrain damper and the
reference tracking of the rotor speed using CPC. Finally, conclusions are drawn in Section 6.

## 2 Iterative feedback tuning theory

This section introduces the IFT method. First, the main details of IFT are given in Section 2.1, followed by the analysis of IFT for systems with a reference input containing an offset in Section 2.3.





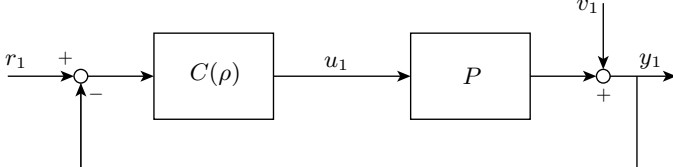

**Figure 1.** Block diagram of a closed-loop system.

## 2.1 IFT introduction

The basic rationale of IFT (Hjalmarsson et al., 1998; Hjalmarsson, 2002) is to minimize a cost function, given for instance by the following quadratic expression

$$J(\rho) = \frac{1}{2N} \sum_{k=1}^{N} \mathrm{E}\left[(y(k,\rho) - r(k))^2 + \lambda u(k,\rho)^2\right], \tag{1}$$

in an iterative manner. The cost function $J(\rho)$ in (1), depends on the (tunable) controller parameters $\rho$, the squared error between the output $y(k,\rho)$ and reference input $r(k)$, and on the squared input signal of the system $u(k,\rho)$, where $k$ indicates the time instance. The cost function is divided by two times the number $N$ of data samples, and involves the expectation $\mathrm{E}[\cdot]$ due to noise. Using a gradient search of the type

$$\rho_{i+1} = \rho_i - \gamma_i R_i^{-1} \frac{\partial J(\rho_i)}{\partial \rho}, \tag{2}$$

where $i$ is the iteration number, $\partial J(\rho_i)/\partial \rho$ is the gradient of the cost function (1), $R_i$ a positive definite matrix (e.g., the Hessian of (1)), an unconstrained optimization problem is obtained.

     It is clear that minimizing (1) boils down to computing the gradient $\partial J(\rho_i)/\partial \rho$ and Hessian $R_i$ at every iteration. Previous studies (e.g., refer to Hjalmarsson et al. (1998); Hjalmarsson (2002)) have shown that these quantities can be obtained straightforwardly from the closed-loop system, by conducting a number of experiments. To see this, first consider the partial derivative

of the cost function $J(\rho)$ with respect to the controller parameter vector $\rho$

$$\frac{\partial J}{\partial \rho}(\rho) = \frac{1}{N} \sum_{k=1}^{N} \mathrm{E}\left[y(k,\rho)\frac{\partial y}{\partial \rho}(k,\rho) + \lambda u(k,\rho)\frac{\partial u}{\partial \rho}(k,\rho)\right]. \tag{3}$$

which involves the signals $\partial y/\partial \rho(k,\rho)$ and $\partial u/\partial \rho(k,\rho)$. Thus, in each iteration the following is required

     1. The signals $r(k)$, $y(k,\rho_i)$ and $u(k,\rho_i)$;

     2. The gradients $\partial y/\partial \rho(k,\rho_i)$ and $\partial u/\partial \rho(k,\rho_i)$;

3. Unbiased estimates of the products $y(k,\rho_i)\partial y/\partial \rho(k,\rho_i)$ and $u(k,\rho_i)\partial u/\partial \rho(k,\rho_i)$.

The signals of the first requirement can be obtained from running a closed-loop experiment as in Figure 1.





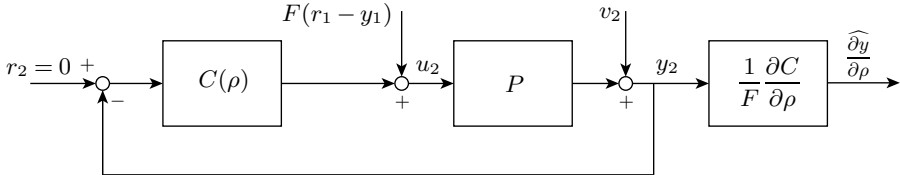

**Figure 2.** Block diagram of the closed-loop gradient experiment.

Obtaining the signals of the latter two requirements are slightly more involved. In order to derive the required gradients and unbiased estimates, consider the block scheme of the closed-loop system in Figure 1. From the block scheme, it is readily observed that

$$y_1(k,\rho) = Pu_1(k,\rho) + v(k), \tag{4}$$

$$u_1(k,\rho) = C(\rho)\left(r_1(k) - y_1(k,\rho)\right). \tag{5}$$

Taking the partial derivatives of the latter signals with respect to the controller parameters $\rho$ gives

$$\frac{\partial y_1}{\partial \rho}(k,\rho) = P\frac{\partial u_1}{\partial \rho}(k,\rho), \tag{6}$$

$$\frac{\partial u_1}{\partial \rho}(k,\rho) = \frac{\partial C}{\partial \rho}(r_1(k) - y_1(k,\rho)) - C(\rho)\frac{\partial y_1}{\partial \rho}(k,\rho). \tag{7}$$

Substitution of (7) in (6) yields

$$\frac{\partial y_1}{\partial \rho}(k,\rho) = (I + PC(\rho))^{-1}P\frac{\partial C}{\partial \rho}(r_1(k) - y_1(k,\rho)) = S(\rho)P\frac{\partial C}{\partial \rho}(r_1(k) - y_1(k,\rho)), \tag{8}$$

where $S(\rho) = (I + PC(\rho))^{-1}$ is the sensitivity function. The gradient in (8) can be obtained by injecting $r_1(k) - y_1(k,\rho)$ at the process input $u_2(k,\rho)$ according to Figure 2. This experiment is the so-called *gradient* experiment. Notice that Figure 2 includes a scaled injection signal with a factor $F$ at the process input as well as a factor $1/F$ at the output, which will become clear in the following paragraphs; for now assume $F = 1$. It should also be noted that the gradient $\widehat{\partial y}/\partial \rho(k,\rho)$ obtained through Figure 2 is an estimate of (8), because it is contaminated with noise $v_2(k)$, i.e.,

$$\frac{\widehat{\partial y}}{\partial \rho}(k,\rho) = \frac{\partial C}{\partial \rho}\left(S(\rho)P(r_1(k) - y_1(k,\rho)) + S(\rho)v_2\right). \tag{9}$$

Finally, note that the subscript indicates the experiment number and should not be confused with the iteration number $i$. Performing the gradient experiment as in Figure 2 avoids the need of an inverse of the controller, which will become clear in the next section.

The input gradient signal $\partial u/\partial \rho(k,\rho)$ can be obtained in a similar way (refer to Hjalmarsson et al. (1998) for a derivation). It can be derived that

$$\frac{\partial u}{\partial \rho}(k,\rho) = (I + PC(\rho))^{-1}\frac{\partial C}{\partial \rho}(r_1(k) - y_1(k,\rho)) = S(\rho)\frac{\partial C}{\partial \rho}(r_1(k) - y_1(k,\rho)), \tag{10}$$



which means that the gradient can be obtained from Figure 2 by multiplying the process input $u_2(k, \rho)$ with $\partial C/\partial \rho$ (or by $1/F \cdot \partial C/\partial \rho$ when the scaling factor $F$ is applied). Again, this gradient is an estimate, because it is contaminated with $v_2(k)$, and is therefore denoted by $\widehat{\partial u}/\partial \rho(k, \rho_i)$.

Under some mild assumptions of the noise properties (zero-mean noise and the noise should be uncorrelated in each experi-
ment), it can be shown (Hjalmarsson, 2002) that $y_1(k, \rho_i)\widehat{\partial y}/\partial \rho(k, \rho_i)$ and $u_1(k, \rho_i)\widehat{\partial u}/\partial \rho(k, \rho_i)$ are unbiased estimates.

The matrix $R$ in (2) is often replaced by an approximation of the Hessian. When $y(k, \rho) - r(k)$ is small, the Gauss-Newton
direction

$$R = \frac{1}{N}\sum_{k=1}^{N}\left[ \frac{\widehat{\partial y}}{\partial \rho}(k, \rho)\frac{\widehat{\partial y}}{\partial \rho}^{\mathrm{T}}(k, \rho) + \lambda \frac{\widehat{\partial u}}{\partial \rho}(k, \rho)\frac{\widehat{\partial u}}{\partial \rho}^{\mathrm{T}}(k, \rho) \right] \tag{11}$$

can be a suitable choice (Hjalmarsson, 2002). However, the approximated Hessian in (11) will be biased because of the dis-
turbances. Typically, this will slow down the convergence of the algorithm. In Solari and Gevers (2004), it is shown that an
unbiased estimate of the Hessian can be obtained on the basis of two additional closed-loop experiments. In this paper, however,
it was found that the approximated Hessian (11) performed sufficiently well and was therefore the preferred choice.

## 2.2 Improving the signal-to-noise ratio

In order to improve the signal-to-noise ratio in the gradient experiment, it is suggested to replace $r_1(k) - y_1(k, \rho)$ by $F(r_1(k) - y_1(k, \rho))$ (Hjalmarsson, 2002) in Figure 2. Consequently, the output $\widehat{\partial y}/\partial \rho(k, \rho)$ should be divided by the scaling factor $1/F$.
The scaling factor $F$ provides a means to influence the signal-to-noise ratio. For a discussion on the optimal choice of $F$, refer
to Hjalmarsson and Gevers (1997).

In the later sections, the gain $F$ needs not only to be used to improve the signal-to-noise ratio, but also to appropriately
scale the input injection signal $r_1(k) - y_1(k, \rho)$. In certain cases, the reference $r(k)$ and/or measured signals $y(k, \rho)$ can be
orders of magnitude larger (or smaller) than the process input $u(k, \rho)$, which means that injection of $r_1(k) - y_1(k, \rho)$ with large
amplitude would lead to an undesired input. Hence, $F$ can be used to scale this signal to the desired input level.

## 2.3 IFT for systems with offset in reference inputs

In the previous subsection, the controller $C$ was subject to IFT where it was assumed that $r(k)$ could be set to zero. In this
subsection, the details of iteratively optimizing controller $C$ when operating at a non-zero reference input $r(k)$ are analyzed.
Specifically, the details in this part are tailored to the case of optimizing for instance the CPC. In the CPC loop, the generator
speed is tried to be held close to the rated generator speed. Thus, the reference input can be used to generate a step response.
Then, the reference $r(k)$ can be written as $r(k) = r_o + r_n(k)$, where $r_o \neq 0$ is the constant offset (e.g., the rated generator speed)
and $r_n$ the reference step. Similar to the previous section, the offset $r_o$ will contaminate the gradient signals and therefore a
third experiment is required to remove this contamination.

The experiments to obtain unbiased gradient estimates are as follows (Hjalmarsson, 2002). First, an experiment is carried
out where a step change $r_1(k) = r_o + r_n(k)$ is applied to the closed-loop system according to Figure 1. Recording the process





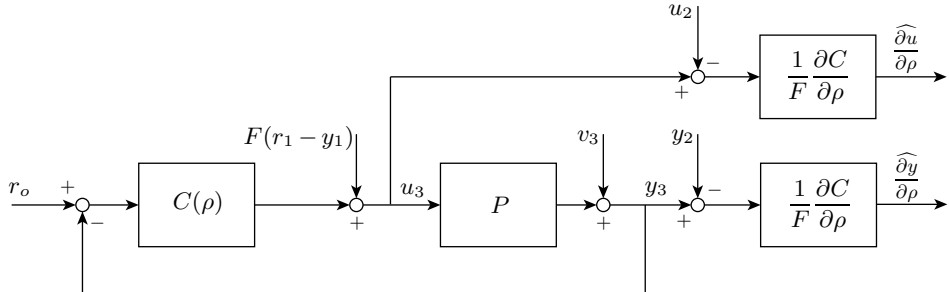

**Figure 3.** Block diagram of modified closed-loop gradient experiment for reference inputs with offsets.

input $u_1(k,\rho)$ and the output $y_1(k,\rho)$

$$y_1(k,\rho) = S(\rho)\big(PC(\rho)(r_o + r_n(k)) + v_1(k)\big), \tag{12}$$

$$u_1(k,\rho) = S(\rho)C(\rho)\big(r_o + r_n(k) - v_1(k)\big), \tag{13}$$

gives the first set of signals. Second, an experiment identical to the first experiment is carried out, except that there is not a step

change in the reference, hence $r_2(k) = r_0$. This yields the second set of signals

$$y_2(k,\rho) = S(\rho)\big(PC(\rho)r_o + v_2(k)\big), \tag{14}$$

$$u_2(k,\rho) = S(\rho)C(\rho)\big(r_o - v_2(k)\big). \tag{15}$$

The third experiment is the experiment where the recorded signals of the first and second experiment are used to obtain an estimate of the gradient. The control configuration for the gradient experiment is shown in Figure 3, where it is seen that the

reference input equals $r_o$. For this configuration, the gradient estimate signals read (also see Appendix A.1)

$$\frac{\widehat{\partial y}}{\partial \rho}(k,\rho) = \frac{1}{F}\frac{\partial C}{\partial \rho}\left(S(\rho)\big[PF(r_1(k) - y_1(k,\rho)) + PC(\rho)r_o + v_3(k)\big] - y_2(k,\rho)\right), \tag{16}$$

$$\frac{\widehat{\partial u}}{\partial \rho}(k,\rho) = \frac{1}{F}\frac{\partial C}{\partial \rho}\left(S(\rho)\big[F(r_1(k) - y_1(k,\rho)) + C(\rho)r_o - C(\rho)v_3(k)\big] - u_2(k,\rho)\right), \tag{17}$$

from which it can be seen that both noise and the reference offset $r_o$ perturb the estimate. Fortunately, the offset $r_o$ in the gradient estimates can be removed by subtracting (14) from (16) and (15) from (17) during the gradient experiment (see Figure 3).

It is noted that the first and second experiment are identical, but are both required in order for $y_1(k,\rho_i)\widehat{\partial y}/\partial\rho(k,\rho_i)$ and $u_1(k,\rho_i)\widehat{\partial u}/\partial\rho(k,\rho_i)$ to be unbiased (Hjalmarsson, 2002) under the same assumptions as before.

### 2.4    IFT for systems with multiple controllers

The IFT method discussed in the previous part cannot directly be applied to control systems where multiple (decoupled) controllers are working on the same input signal. Consider in this case for example the torque controller and the drivetrain

controller. The torque controller regulates the generator torque in such a way that the rotor speed provides maximum power





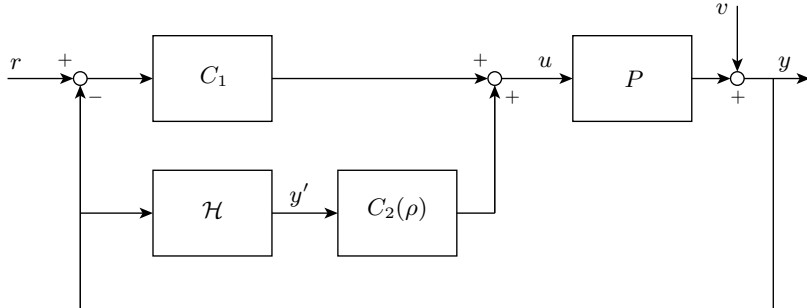

**Figure 4.** Block diagram of the closed-loop system which involves multiple controllers (representing the scenario of torque control and drivetrain damping).

extraction from the wind. At the same time, a drivetrain damper adds a small torque ripple on the regulated generator torque in order to reduce drivetrain oscillations (which will be further explained in Section 4). Thus, in such case the reference signal $r(k)$ in Figure 2 cannot be set to zero. Moreover, the torque and damping controllers might have some interaction such that biased estimates are obtained. In order to apply IFT to this kind of control systems, it is shown that a similar procedure as for the case

with reference offsets can be used.

The basic control system analyzed in this section is shown in Figure 4. The control system consists of controllers $C_1$ and $C_2(\rho)$, where the latter is subject to optimization. Controller $C_1$ works on the error between a reference $r(k)$ and the measured output $y(k,\rho)$, while $C_2(\rho)$ works on the output $y(k,\rho)$ filtered by $\mathcal{H}(s)$. The role of the high-pass filter will become clear at a later point. The controller $C_2(\rho)$ is without loss of generality implemented with positive feedback.

The first experiment is identical to the single controller case as described in the previous paragraphs: the closed-loop system in Figure 4 is used to obtain the following signals

$$y_1(k,\rho) = S(\rho)\big(PC_1 r_1(k) + v_1(k)\big), \tag{18}$$

$$u_1(k,\rho) = S(\rho)\big(C_1 r_1(k) - C_1 v_1(k) + C_2(\rho)\mathcal{H}v_1(k)\big), \tag{19}$$

where $S = (I + PC_1 - PC_2(\rho)\mathcal{H})^{-1}$ is the sensitivity function. The signals (18)-(19) determine the cost function given in (1).

The gradient signal related to the output $y_1(k,\rho)$ is obtained as follows. First note that

$$y(k,\rho) = Pu(k,\rho) + v(k) \tag{20}$$

such that the partial derivative thereof is given by

$$\frac{\partial y}{\partial \rho}(k,\rho) = P\frac{\partial u}{\partial \rho}(k,\rho). \tag{21}$$

Then, note that the process input $u(k,\rho)$ in Figure 4

$$u(k,\rho) = C_1(r(k) - y(k,\rho)) + C_2(\rho))\mathcal{H}y(k,\rho) \tag{22}$$



has partial derivative

$$\frac{\partial u}{\partial \rho}(k,\rho) = -C_1\frac{\partial y}{\partial \rho}(k,\rho) + \frac{\partial C_2}{\partial \rho}\mathcal{H}y(k,\rho) + C_2(\rho)\mathcal{H}\frac{\partial y}{\partial \rho}(k,\rho). \tag{23}$$

Substituting (23) in (21) and manipulating, yields

$$\frac{\partial y}{\partial \rho}(k,\rho) = S(\rho)P\frac{\partial C_2}{\partial \rho}\mathcal{H}y(k,\rho). \tag{24}$$

Similarly, the gradient related to the input can be found to be (see Appendix A.2)

$$\frac{\partial u}{\partial \rho}(k,\rho) = S(\rho)\frac{\partial C_2}{\partial \rho}(\rho)\mathcal{H}y(k,\rho). \tag{25}$$

Thus, the signals (24)-(25) can be obtained by injecting the high-pass filtered output $\mathcal{H}y(k,\rho)$ or simply $y'(k,\rho)$ at the process input $u(k,\rho)$ in a gradient experiment according to Figure 5. Doing so, one then obtains the following gradient estimates

$$\widehat{\frac{\partial y}{\partial \rho}}(k,\rho) = \frac{1}{F}\frac{\partial C_2}{\partial \rho}\bigg(S(\rho)\big[PC_1r(k) + FP\mathcal{H}y_1(k,\rho) + v_3(k)\big] - y_2(k,\rho)\bigg), \tag{26}$$

$$\widehat{\frac{\partial u}{\partial \rho}}(k,\rho) = \frac{1}{F}\frac{\partial C_2}{\partial \rho}\bigg(S(\rho)\big[C_1r(k) - C_1v_3(k) + C_2(\rho)\mathcal{H}v_3(k) + F\mathcal{H}y_1(k,\rho)\big] - u_2(k,\rho)\bigg). \tag{27}$$

Comparing the above equations with (24)-(25) and it can be observed that the estimates are biased. Now, by running an experiment identical to the first experiment, one acquires $y_2(k,\rho)$ and $u_2(k,\rho)$ identical to (18)-(19). Subtracting these, as shown in Figure 5, cancels the undesired terms in the gradient signals (26)-(27) such that the desired gradient is obtained which is only perturbed by noise terms.

The high-pass filter $\mathcal{H}$ is incorporated for practical consideration. In the gradient experiment, the signal $F\mathcal{H}y_1(k,\rho)$ is injected in the system. In the case of drivetrain damping, this would mean to inject the measured generator speed $y_1(k,\rho)$ with a scaling factor $F$. As the generator speed during operating is larger than zero, this would imply to insert a step change in the demanded generator torque. High-pass filtering the measured speed causes to inject a signal that varies around zero.

## 2.5  Including cost function weights

This section shortly discusses weighting filters in the cost function. The cost function in (1) can be modified to include time and frequency weights. First, consider the following so-called zero-weighting mask of the cost function (Lequin et al., 1999, 2003)

$$J(\rho) = \frac{1}{2N}\sum_{k=N_0}^{N} \mathrm{E}\left[(y(k,\rho) - r(k))^2 + \lambda u(k,\rho)^2\right], \tag{28}$$

where the index of the summation starts at $N_0 > 1$. The motivation for this mask, often used for step response tuning, is
as follows. Typically, the main objective of step response tuning is to move the system quickly from one point to another. The settling time is an important parameter in this context. Without zero-weighting mask, the controller is tuned such that the reference is as close as possible matched, while in practice one does not care too much about the trajectory to the new reference





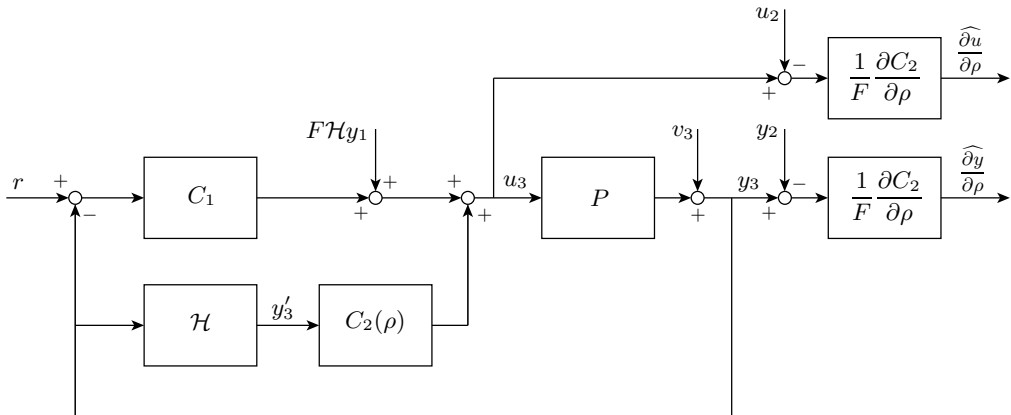

**Figure 5.** Block diagram of the gradient experiment for the multiple controller scenario. The controller $C_1$ is fixed and controller $C_2(\rho)$ is subject to IFT. The signals $\mathcal{H}y_1(k,\rho)$, $y_2(k,\rho)$ and $u_2(k,\rho)$ are obtained from two others experiments and are required to obtain unbiased gradient estimates.

value, as long as the overshoot is not too large. By zero-weighting the transient trajectory in the cost function, the algorithm tries to achieve a fast settling time.

The cost function can also incorporate frequency weights $L_y$ and $L_u$

$$J(\rho) = \frac{1}{2N} \sum_{k=N_0}^{N} \mathrm{E}\left[ (L_y(y(k,\rho) - r(k)))^2 + \lambda(uL_u(k,\rho))^2 \right], \tag{29}$$

which filter the error $y(k,\rho) - r(k)$ and input $u(k)$ accordingly. With the frequency weights one can emphasize or suppress frequency bands in the cost function. For example, in the case of drivetrain damping, the measured output signal will not only be composed of the drivetrain resonance frequency, but also many other disturbances with various frequency components. Thus, filtering the drivetrain frequencies in the cost function makes sure the controller will focus on the filtered frequencies.

The cost function (29) requires filtering of $y(k,\rho) - r(k)$ and $u(k,\rho)$ with $L_y$ and $L_u$ respectively. Moreover, the derivative of (29) with respect to the control parameters

$$\frac{\partial J}{\partial \rho}(\rho) = \frac{1}{N} \sum_{k=1}^{N} \mathrm{E}\left[ L_y y(k,\rho) L_y \frac{\partial y}{\partial \rho}(k,\rho) + \lambda L_u u(k,\rho) L_u \frac{\partial u}{\partial \rho}(k,\rho) \right], \tag{30}$$

then involves the filters $L_y$ and $L_u$ as well. Thus, this also requires the gradient signals need to be passed through the frequency filters.

## 2.6 Stability

Finally, this section is ended with a note on stability. The iterative optimization of the controller parameters in $\rho$ described in the previous paragraphs can have robustness issues. This is caused by the fact that, typically, there are no guarantees that the controller remains stable during the iteration. For that reason a number of articles (Heertjes et al., 2016; Procházka et al., 2005;





**Table 1.** NREL 5MW wind turbine description (Bossanyi and Witcher, 2009)

| Description | Value |
| --- | --- |
| Rated power | $5\,\mathrm{MW}$ |
| Number of blades | 3 |
| Rotor diameter | $126\,\mathrm{m}$ |
| Orientation | Upwind |
| Hub height | $90\,\mathrm{m}$ |
| Gearbox ratio | 97 |
| Rated wind speed | $11.3\,\mathrm{m\,s^{-1}}$ |
| Rated rotor speed | $12.1\,\mathrm{rpm}$ |
| Rated generator torque | $43.093\,\mathrm{kNm}$ |
| Drivetrain natural frequency | $10.49\,\mathrm{rad\,s^{-1}}$ |

de Bruyne and Kammer, 1999; Veres and Hjalmarsson, 2002) have appeared on the subject of including stability constraints for the optimization procedure. These algorithms are not considered here, because the authors have not encountered any stability issues.

## 3   Wind turbine and simulation environment

In this study, the NREL 5MW reference wind turbine model is used. The model does not represent an existing wind turbine, but is considered to reflect typical commercial wind turbines of similar ratings. The turbine has three blades in an upwind rotor configuration, a rotor diameter of $126\,\mathrm{m}$, and reaches the rated power output of $5\,\mathrm{MW}$ at a wind speed of $12.1\,\mathrm{m\,s^{-1}}$. A summary of the most important parameters is listed in Table 1. The wind turbine is a variable-speed variable-pitch machine, such that the rotor speed in below-rated wind is regulated by means of the generator torque, and in above-rated wind by collectively

pitching the blades. The natural frequency of the drivetrain dynamics, which is important for the drivetrain damper that will be designed in Section 4, is at $\omega_r = 10.49\,\mathrm{rad\,s^{-1}}$.

The relevant control scheme for the scope of this study is given in Figure 6. The measured generator speed $\Omega_{\mathrm{gen}}$ forms the input for the controllers. The torque controller provides a demanded generator torque setpoint $T_{\mathrm{trq}}$ to regulate the rotor speed in below-rated wind and drivetrain damping can be achieved by superimposing $T_{\mathrm{trq}}$ with a small torque ripple $T_{\mathrm{dtd}}$,

such that the final demanded generator torque is given by $T_{\mathrm{gen}}$. In above-rated conditions, the generator speed is held constant at $\Omega_{\mathrm{ref}} = \Omega_{\mathrm{rated}}$ by means of collective blade pitch setpoints $(\theta_1, \theta_2)$ generated by the CPC. The drivetrain damper and the CPC will be optimized using IFT in Section 4 and 5.

The software package GH Bladed $4.20$ (Garrad Hassan; Garrad Hassan & Partners Ltd, a, b), which is a certified and widely used wind turbine design software package in industry, is used to simulate the behavior of the wind turbine in response to

a supplied wind field. To do so, the structural model of the turbine is modeled by a multi-body approach combined with a





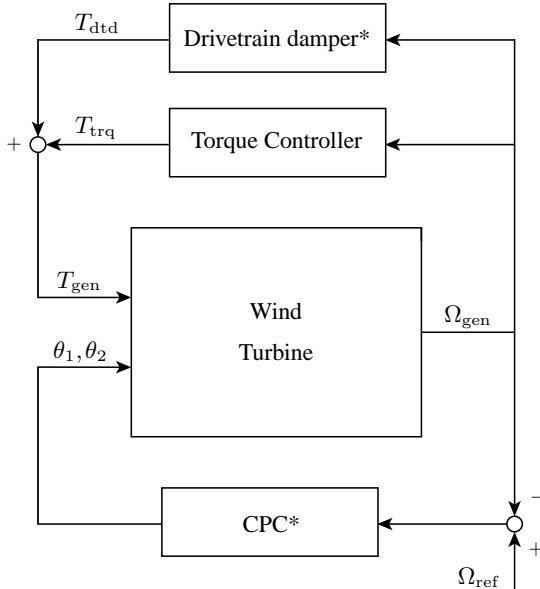

**Figure 6.** Overview of the relevant controller configuration in this study. The starred controllers will be the subject for IFT.

modal representation of the flexible components. The rotor aerodynamics are modeled by combining blade and momentum theory. Moreover, Bladed allows for testing of new control algorithms by compiling an external controller designed in e.g., MATLAB Simulink (Mathworks) to a DLL file (Houtzager). The DLL file is then used during the calculations to obtain the closed-loop dynamics of the wind turbine. The controllers are discretized using the Tustin approximation and run at a sampling

time of $0.01\,\mathrm{s}$. The resulting turbine response is written to data files, which can be post-processed in Bladed or external software to analyze the response. The data in the log files is recorded with a sampling time of $0.05\,\mathrm{s}$. The interested reader is referred to the theory manual of Bladed (Garrad Hassan & Partners Ltd, b) for details on the calculation methods.

## 4  IFT of a drivetrain damper

Wind turbines which have a geared drivetrain are known to have a lightly damped drivetrain mode. Subjected to a turbulent

wind, the rotor speed will vary despite the speed regulation by torque control and CPC. The rotor speed variations cause the drivetrain mode to be excited, which can lead to heavy oscillations in the drivetrain. In order to prevent this, typically a drivetrain damping controller is included in the control system (e.g., see Figure 6). This controller basically adds a small torque ripple at the drivetrain frequency (Bossanyi, 2000) to the demanded generator torque of the torque controller. Doing so, will dramatically reduce the drivetrain oscillations. Several studies in the past have considered the design of drivetrain

damping control, e.g., refer to (Bossanyi, 2000; Dixit and Suryanarayanan, 2005; Fleming et al., 2011; Wright et al., 2011; Fleming et al., 2013).



In this section, the IFT algorithm c.f. Section 2 is applied to the optimization of the drivetrain damper parameters. In the next paragraph, the controller parameterization is given as well as the classical design approach. Then, a parameter study is carried out to visualize the cost function. Subsequently, the controller is iteratively optimized for realistic loading scenarios as well as for different algorithm settings. Finally, the results of this case study are discussed.

## 4.1 Baseline damping controller design

Typically, the drivetrain controller is chosen to be a bandpass filter of the form (Bossanyi, 2000; Bossanyi and Witcher, 2009)

$$C_{\mathrm{dtd}}(s) = \frac{2K\omega\zeta s}{s^2 + 2\zeta\omega s + \omega^2}, \tag{31}$$

where $K$ is the bandpass gain, $\omega$ is the drivetrain frequency and $\zeta$ influences the damping. With a (linearized) model of the drivetrain dynamics $G_{\mathrm{dtd}}$, this bandpass filter can be tuned to damp the drivetrain oscillations. To this end, a linearized model of the drivetrain dynamics is obtained from Bladed using the built in linearization tool. With this model the drivetrain damper is designed using classical loopshaping techniques (Skogestad and Postlethwaite, 2006). The resulting Bode diagrams of the drivetrain dynamics, the open-loop (controller times drivetrain dynamics) and the closed-loop are shown in Figure 7. From the open-loop dynamics shown in Bode diagram, it is recognized that the bandpass gain $K$ can be increased infinitely (i.e., because the phase plot never crosses $-180°$). Hence, during the iterative optimization the controller cannot become unstable (for positive values of the controller parameters). The bandpass filter which is used in Figure 7 has $K = 2500$, $\zeta = 0.3$, and $\omega = \omega_d = 10.49\,\mathrm{rad\,s}^{-1}$ and is considered as the baseline controller. This filter has been experimentally verified to yield satisfactory performance.

## 4.2 Parameter study for drivetrain damping

Before the IFT algorithm was implemented, first a parameter study using the linearized drivetrain dynamics was carried out. In order to gain understanding of the optimization problem, the closed-loop system including the drivetrain dynamics $G_{\mathrm{dtd}}$ and the bandpass filter $C_{\mathrm{dtd}}$ is simulated for a range of parameters $(K, \omega, \zeta)$. A disturbance signal at the drivetrain frequency is injected at the output. The cost function is taken as (1), i.e., both the output and the input are weighted. The cost function for combinations of $K$ and $\omega$ is shown in Figure 8. The results indicate that the cost function has a large area where it is almost optimal and which becomes wider for increasing $\zeta$. This means that for this problem many combinations of $K$ and $\omega$ exists which give almost identical results. Moreover, this also indicates that the parameters are likely to quickly converge to almost minimum cost values, but will slowly converge to the optimum value. It also shows that the baseline damping filter with $\zeta = 0.3$ could be improved (at least for the simulation case) by decreasing the damping $\zeta$ of the filter. Finally, the cost function plot shows that the phase of the controlled system can be adjusted by increasing/decreasing the frequency $\omega$ and at the same time increasing/decreasing the bandpass gain $K$ to maintain practically the same compensation performance.




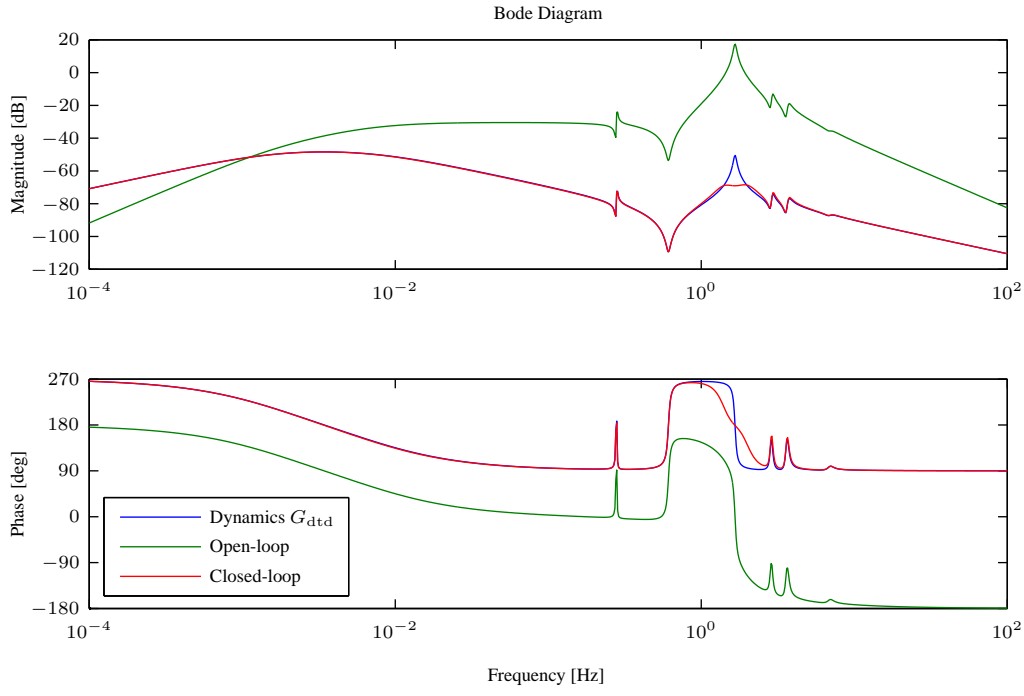

**Figure 7.** Bode diagrams of the drivetrain dynamics, the open-loop (controller times drivetrain dynamics), and the closed-loop system.

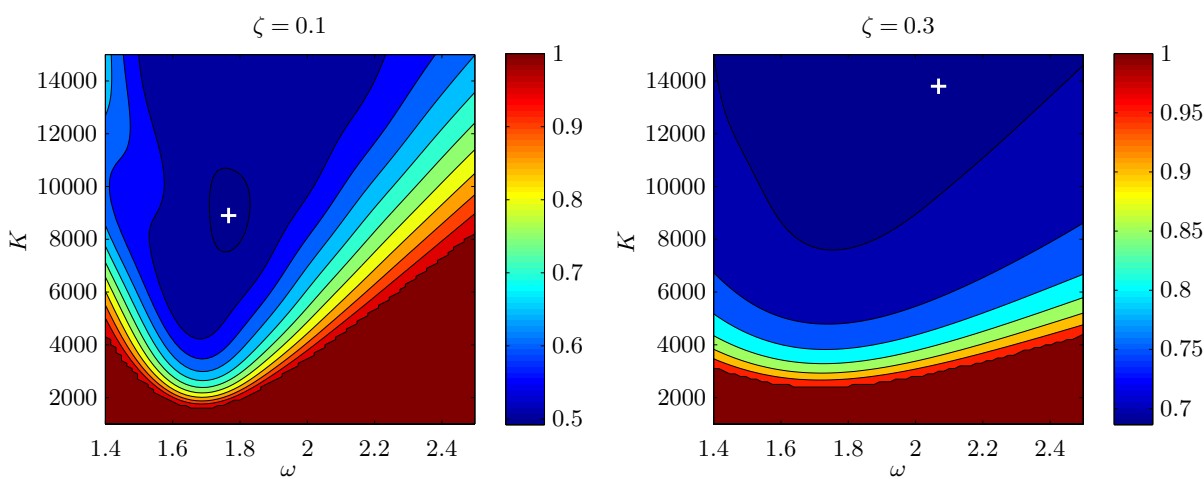

**Figure 8.** Close-up of the optimal parameter combinations for the drivetrain excitation problem with $\zeta = 0.1$ (left) and $\zeta = 0.3$ (right). The results are thresholded and normalized for clarity. The results show large areas where the cost function is almost optimal. The minimum values of the cost function for each case is indicated by the white + marker.



### 4.3 IFT of drivetrain damping

The input of the drivetrain damping controller (31) is taken to be the measured generator speed. The generator speed signal consists of many frequency components arising from the interaction between the wind turbine system and the wind. For drivetrain damping one is only interested in the frequencies close to the drivetrain frequency. Hence, the controller scheme is taken to be identical to Figure 4, where $C_1$ is the torque controller and $C_2$ the drivetrain damper. The input to the system is $u = T_{\text{gen}}$ and the output $y = \Omega_{\text{gen}}$. The measured generator speed signal $\Omega_{\text{gen}}$ is passed through the high-pass filter

$$\mathcal{H}(s) = \frac{s^2}{s^2 + 2\zeta_{\mathcal{H}}\omega_{\mathcal{H}}s + \omega_{\mathcal{H}}^2}, \tag{32}$$

where $\omega_{\mathcal{H}} = 0.63\,\text{rad s}^{-1}$ is the cut-off frequency and $\zeta_{\text{H}} = 0.7$ determines the damping of the filter. In order to make sure that the cost function is dominated by the drivetrain excitations, the output component of the cost function (1) is filtered. The weighting filter

$$L_y(s) = \frac{\omega_r s}{s^2 + 2\zeta_r\omega_r s + \omega_r^2}, \tag{33}$$

with $\omega_r = 10.49\,\text{rad s}^{-1}$ at the drivetrain frequency and $\zeta_r = 0.1$, is effectively an inverted notch filter passing through the frequencies close to the drivetrain frequency and attenuating other frequencies. The input component of the cost function is also filtered, as one in this case is only interested in the 'high' frequency part of the generator torque setpoint $T_{\text{gen}}$. Therefore, the input component is also filtered with the high-pass filter (32). The cost function then becomes

$$J_{\text{dtd}}(\rho) = \frac{1}{2N}\sum_{k=1}^{N}\text{E}\left[(L_y\mathcal{H}\Omega_{\text{gen}}(k,\rho))^2 + \lambda\mathcal{H}T_{\text{gen}}(k,\rho)^2\right], \tag{34}$$

which thus consists of the measured response of the generator speed $\Omega_{\text{gen}}$ filtered by (32) and (33) and the demanded generator torque $T_{\text{gen}}$ filtered by (32). These filters are also included in the gradient experiment configuration in Figure 5 just before the controller derivatives.

The experiments of IFT for the iterative drivetrain controller optimization are according to Section 2.4 as follows:

**First experiment** ($T : 0 - 20\,\text{s}$)  In the first experiment, the closed-loop system is operated for $20\,\text{s}$ (i.e., $N = 2000$ samples), with $C_1$ and $C_2$ identical to Figure 4. The output $\Omega_{\text{gen}}$ is filtered by (32) and (33), and the input $T_{\text{gen}}$ is filtered by (32).

**Second experiment** ($T : 20 - 40\,\text{s}$)  The second experiment is identical to the first experiment.

**Third experiment** ($T : 40 - 60\,\text{s}$)  In the third experiment, the filtered output $\mathcal{H}\Omega_{\text{gen}}$ is added to the torque setpoint $T_{\text{gen}}$ and the recorded signals from the second experiment are subtracted as shown in Figure 5.

After the third experiment at $T = 63\,\text{s}$, the controller parameters are updated and subsequently the next $7\,s$ are used for transients, due to the controller parameter update, to disappear before the next iteration is started. Thus, a full iteration takes $70\,s$ seconds. In the results the IFT algorithm is also adjusted to collect $N = 1000$ and $N = 3000$ per experiment. In these cases the





experiment lengths are adjusted accordingly and the iterations therefore in total take $40\,\mathrm{s}$ and $100\,\mathrm{s}$, respectively. In all cases, the first iteration starts at $T = 30\,\mathrm{s}$ after the start of the simulation such that all initial transients are disappeared.

The generator torque setpoint $T_{\mathrm{gen}}$ provided by the IFT algorithm is limited to $\pm 1.8\,\mathrm{kNm}$. Moreover, the generator torque setpoint is also limited in the rate of change, i.e., a maximum rate of $\pm 20\,\mathrm{kNm\,s^{-1}}$ is allowed.

## 4.4 Results

The results for IFT of the drivetrain damper are subdivided into four parts each covering a different aspect of the IFT results.

### 4.4.1 General analysis of results

In the first part, the results for a number of general settings of IFT are analyzed. The wind field considered here has a mean wind speed of $14\,\mathrm{m\,s^{-1}}$ and a turbulence intensity of $10\,\%$. The number of data points considered is $N = 2000$ corresponding to $20\,\mathrm{s}$ of simulated time (recall that the controllers run at a sampling time of $0.01\,\mathrm{s}$) and the parameter update step size of (2) is set to $\gamma = 0.3$. The adjustable signal-to-noise ratio parameter $F$ is set to 2000, which was experimentally found to provide a good trade-off between the signal-to-noise ratio and the amplitude of the injected signal.

First, the convergence of the cost function and controller parameters for different optimization cases are considered. In total four cases are considered:

**Baseline** In the baseline case the controller parameters are held constant and is given for reference;

$\rho_{\mathrm{base}}$ In the second case, the initial parameters $\rho_{0,\mathrm{base}}$ are equal to the baseline parameters (i.e., $K = 2500$, $\omega = 10.49\,\mathrm{rad\,s^{-1}}$, and $\zeta = 0.3$), from which IFT procedure is started. The input weighting is chosen to be $\lambda = 5 \cdot 10^{-7}$;

$\rho_{\mathrm{base},\lambda}$ The third case is identical to the second case except that the input weighting is smaller, i.e., $\lambda = 2 \cdot 10^{-7}$;

$\rho_{\mathrm{subopt}}$ In the final case, the initial parameters $\rho_{0,\mathrm{subopt}}$ are chosen to be far from the baseline case: $K = 1000$, $\omega = 6.28\,\mathrm{rad\,s^{-1}}$, and $\zeta = 0.4$. The input weighting is identical to the second case.

The results of the comparison are shown in Figure 9. From the three plots related to the cost, it is observed that the baseline controller is already rather close to optimal controller performance. The controller with suboptimal initial controller parameters converges towards the other controller cases in roughly 10 iterations. The main performance improvement is observed in the generator torque effort. It can be observed that the bandpass gain $K$ and bandpass damping $\zeta$ of the baseline controller should, respectively, be increased and decreased in order to improve the performance. The bandpass frequency $\omega$ remains close to the drivetrain resonance frequency $\omega_d$ during the optimization. The influence of the lower input weighting $\lambda$ in the case of $\rho_{\mathrm{base},\lambda}$ is clearly seen in the bandpass trajectory.

With the controller parameters obtained in the last iteration of the second case, a comparison is made with the baseline case in a normal design load case according to IEC (2005). The frequency spectra of the demanded generator torque $T_{\mathrm{gen}}$ and the resulting generator speed $\Omega_{\mathrm{gen}}$ are shown in Figure 10. From the spectrum plots it can be observed that the IFT optimized controller yields a higher damping around the drivetrain frequency $\omega_d$. The optimized parameterization also slightly increases



the frequency contents around the drivetrain mode. The demanded generator torque $T_{\text{gen}}$ displays a similar result. There is slightly more energy concentrated at and around the drivetrain frequency, while at low and high frequencies the energy has reduced.

### 4.4.2 Impact of $F$ on results

In this paragraph, the influence of the scaling factor $F$ on the IFT performance is investigated. To this purpose, the IFT algorithm is carried out on the drivetrain damper using different values of $F$. The initial parameters are identical to the baseline and the step size is chosen to be $\gamma = 0.3$ and the input weight to $\lambda = 5 \cdot 10^{-7}$. The number of data samples collected is again set to $N = 2000$ corresponding to experiment lengths of $20\,\text{s}$. The turbulent wind field is identical as in the previous paragraphs. The results for three cases where F is varied between $F = 1000$ and $F = 2000$ are shown in Figure 11.

The first thing that can be observed from the cost function plot is that the cost function slightly reduces with increasing the scaling factor $F$. The effect of $F$ on the optimization is more recognized in the convergence trajectories of the bandpass gain $K$ and the bandpass damping coefficient $\zeta$. Clearly, by increasing the scaling factor $F$, the convergence rate increases. This is also what one could expect from reasoning, because the excitation of the system increases with $F$, as can also be seen from the lower plot in Figure 11. The choice of $F$ seems to be independent, at least for the considered values, of the final values to

which the controller parameters converge. Thus, when applying IFT, the scaling factor should be carefully chosen such that the choice for $F$ in that sense becomes a tradeoff between the convergence rate and maximum allowed magnitude of the injected signal.

### 4.4.3 Varying experiment length $N$

In the previous results, the length of the experiments in each iteration was $N = 2000$ ($20\,\text{s}$). Here, the iterative optimization

results are compared for experiment lengths of $10\,\text{s}$, $20\,\text{s}$ and $30\,\text{s}$. The step size and input weight are set to $\gamma = 0.3$ and $\lambda = 5 \cdot 10^{-7}$, and the scaling factor is set to $F = 2000$. The results are shown in Figure 12.

It is observed that the experiment length has a clear influence on the variance of obtained results, both for the cost function plots as well as the controller parameters. The results also indicate that for the drivetrain damping case the experiment lengths do not dramatically change the optimization outcomes. It seems that for the $N = 1000$ case the load reduction performance is slightly better than in the other cases. Similarly, the $N = 3000$ case seems to result in the lowest generator torque effort to

reduce the drivetrain oscillations. Moreover, for this case the parameters remain rather close to the original baseline parameters. It can be argued that the $N = 2000$ case provides the best tradeoff between the variance, performance, and convergence time.

### 4.4.4 Varying wind conditions

In the final part of this case study, IFT is also applied to a wind speed at below-rated operating conditions. For comparison,

the $14\,\text{m}\,\text{s}^{-1}$ wind case is also shown. The step size and input weight are once more set to $\gamma = 0.3$ and $\lambda = 5 \cdot 10^{-7}$, the scaling



**Figure 9.** Comparison of three drivetrain torsional damping controllers of which two are subject to IFT. The baseline controller parameters are kept constant and the result is shown as reference. The other three cases involve IFT where the torsional damper is optimized starting from different initial conditions. Results shown are obtained with $\gamma = 0.3$, $F = 2000$, and turbulent wind with mean speed $14\,\mathrm{m\,s^{-1}}$ and $10\,\%$ turbulence intensity.





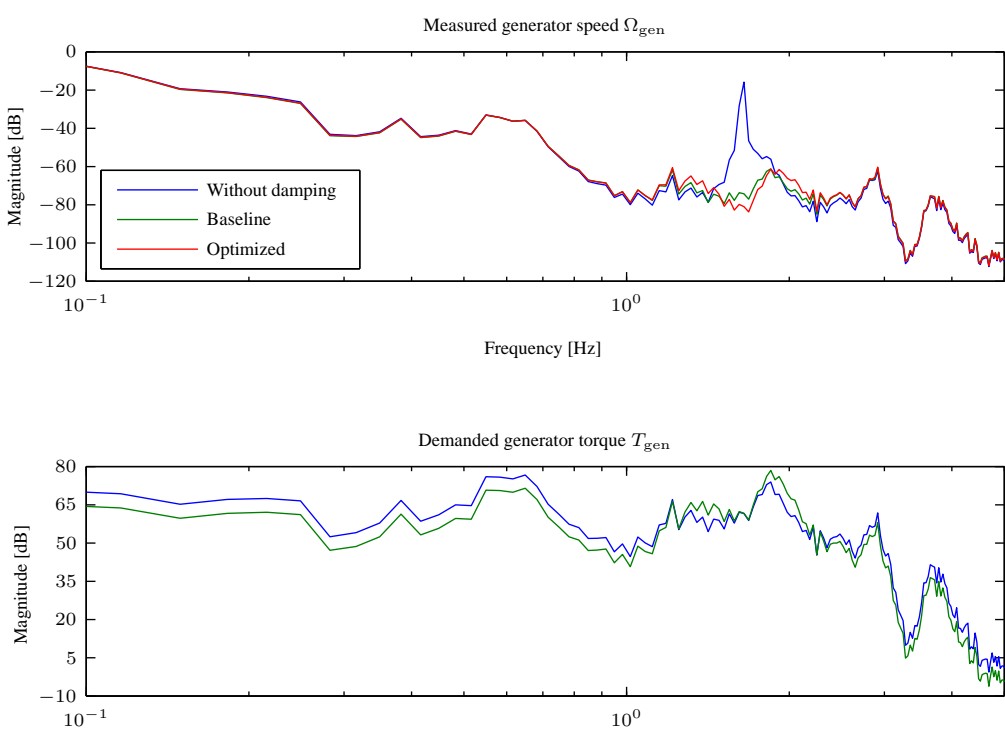

**Figure 10.** Comparison of the baseline and IFT optimized drivetrain controller performance.

factor is chosen to be $F = 3000$ for the $10\,\mathrm{m\,s^{-1}}$ wind and $F = 2000$ for the $14\,\mathrm{m\,s^{-1}}$ wind. The experiment length is kept at $N = 2000$ samples. The results are displayed in Figure 13.

It is observed in the plots that the convergence trajectories for both wind speeds are very similar. The $14\,\mathrm{m\,s^{-1}}$ turbulent wind case excites the drivetrain mode more than the $10\,\mathrm{m\,s^{-1}}$ wind case as is suggested by the increased cost, which this also

5 requires a higher input energy. The cost function plots show a number of iterations with a clearly higher input cost. This is caused by sudden wind speed changes to which the torque controller responds. Although the input in the cost function is high-pass filtered, these sudden changes remain in the input signal. At such occasions, the controller parameters also display relatively large changes. This could be overcome by increasing the cut-off frequency of the high-pass filter that is used to filter the input signal in the cost function, or by increasing the experiment length, such that these effects are averaged (i.e., the

10 experiment length acts as low-pass filter on the results).





**Figure 11.** Comparison of three different signal-to-noise ratio scaling factors $F$. Results are shown for $\gamma = 0.3$, $\lambda = 5 \cdot 10^{-7}$, and turbulent wind with mean speed $14\,\mathrm{m\,s^{-1}}$ and $10\,\%$ turbulence intensity.







**Figure 12.** Comparison of IFT performance for different number $N$ of collected data samples. Result shown are obtained with $\gamma = 0.3$, $\lambda = 5 \cdot 10^{-7}$, $F = 2000$, and turbulent wind with mean speed $14\,\mathrm{m\,s^{-1}}$ and $10\,\%$ turbulence intensity..







**Figure 13.** Comparison of IFT performance for turbulent wind with mean wind speeds $10\,\mathrm{m\,s^{-1}}$ and $14\,\mathrm{m\,s^{-1}}$, both with $10\,\%$ turbulence intensity. Results shown are obtained with $\gamma = 0.3$ and $\lambda = 5 \cdot 10^{-7}$.



**Table 2.** Filter parameters of the different controller components.

| Description | Symbol | Value |
|---|---|---|
| Low-pass filter | | |
| Low-pass filter frequency | $\omega_{\mathrm{LP}}$ | $10.05\,\mathrm{rad\,s}^{-1}$ |
| Low-pass filter damping | $\zeta_{p,\mathrm{LP}}$ | 0.7 |
| Notch filter at 3P frequency | | |
| Notch filter frequency | $\omega_{\mathrm{3P}}$ | $3.77\,\mathrm{rad\,s}^{-1}$ |
| Notch filter damping zero | $\zeta_{z,\mathrm{3LP}}$ | 0.0015 |
| Notch filter damping pole | $\zeta_{p,\mathrm{3LP}}$ | 0.15 |
| Notch filter at drivetrain frequency | | |
| Notch filter frequency | $\omega_d$ | $10.49\,\mathrm{rad\,s}^{-1}$ |
| Notch filter damping zero | $\zeta_{z,d}$ | 0.002 |
| Notch filter damping pole | $\zeta_{p,d}$ | 0.2 |

## 5 IFT of CPC

This section presents the results of the IFT algorithm applied to step tuning of the CPC. In the next paragraph, first the controller structure is given including the details of the IFT method thereof. In the subsequent section the optimization results for several cases are discussed.

### 5.1 CPC design and IFT implementation

The CPC of a wind turbine generally consists of a Proportional Integral (PI) controller cascaded with some filters that prevent from undesired contributions in the pitch signal. The full controller of the CPC loop is given by

$$C_{\mathrm{cpc}} = \left( K_p + \frac{K_i}{s} \right) \times \underbrace{\frac{s^2 + 2\zeta_{z,\mathrm{3P}}\omega_{\mathrm{3P}}s + \omega_{\mathrm{3P}}^2}{s^2 + 2\zeta_{p,\mathrm{3P}}\omega_{\mathrm{3P}}s + \omega_{\mathrm{3P}}^2}}_{\text{Notch at 3P frequency}} \times \underbrace{\frac{s^2 + 2\zeta_{z,d}\omega_d s + \omega_d^2}{s^2 + 2\zeta_{p,d}\omega_d s + \omega_d^2}}_{\text{Notch at dtr frequency}} \times \underbrace{\frac{\omega_{\mathrm{LP}}^2}{s^2 + 2\zeta_{p,\mathrm{LP}}\omega_{\mathrm{LP}}s + \omega_{\mathrm{LP}}^2}}_{\text{Low−pass filter}}, \tag{35}$$

where the filter coefficients are listed in Table 2. The CPC thus includes a notch filter that prevents from working on the 3P

frequency present in the generator speed signal, similarly a notch filter that prevents from reacting to the drivetrain frequency component, and a low-pass filter removing all frequencies above $1.6\,\mathrm{Hz}$. The controller takes as input signal the measured generator speed $\Omega_{\mathrm{gen}}$ and outputs a demanded collective pitch signal $\theta = \theta_1 = \theta_2$, see Figure 6.

The IFT algorithm is applied so as to optimize the step response tracking of the controller in (35). This is achieved by imposing a step change in the rated generator speed signal, i.e., $\Omega_{\mathrm{ref}} = \Omega_{\mathrm{rated}} - \Omega_{\mathrm{step}}$. A negative step change $\Omega_{\mathrm{step}}$ is applied

to prevent the turbine from going into overspeed. The generator speed signal has a constant offset $r_o = \Omega_{\mathrm{rated}}$ and therefore the experiments according to Section 2.3, and Figure 1 and 3 are used:



**First experiment** ($T : 20 - 40\,\mathrm{s}$)  In the first experiment, the step change $\Omega_{\mathrm{ref}} = \Omega_{\mathrm{rated}} - \Omega_{\mathrm{step}}$ in the reference input is applied for $20\,\mathrm{s}$. The generator speed response $\Omega_{\mathrm{gen}}$ and the high-pass-filtered process input $\theta$ are recorded.

**Second experiment** ($T : 0 - 20\,\mathrm{s}$)  In the second experiment, the closed-loop system is operated with $\Omega_{\mathrm{ref}} = \Omega_{\mathrm{rated}}$. The output $\Omega_{\mathrm{gen}}$ and process input $\theta$ are recorded and used in the next experiment.

**Third experiment** ($T : 65 - 85\,\mathrm{s}$)  In the third experiment, the gradients are obtained by operating the closed-loop system with $\Omega_{\mathrm{ref}} = \Omega_{\mathrm{rated}}$, injecting the error signal $F(\Omega_{\mathrm{ref}} - \Omega_{\mathrm{gen}})$ from the previous experiment at the process input $\theta$, using the recorded signals of the first experiment at the process input of the system according to Section 2.3, and filtering with controller derivatives (including the high-pass filter for the input gradient).

The time between the second and third experiment is required to make sure all oscillations due to the step change have
disappeared. At $T = 86\,\mathrm{s}$ the controller parameters are updated. Then, after $34\,\mathrm{s}$ during which the transients due to the gradient experiment and the controller update have disappeared, at $T = 120\,\mathrm{s}$, the next iteration is started. The first iteration starts at $T = 30\,\mathrm{s}$. During the optimization, the maximum pitch rate $\dot{\theta}$ is limited to $\pm 8^{\circ}\,\mathrm{s}^{-1}$. Notice that the order of the first and second experiment have been reversed during the optimization in comparison to Section 2.3.

The cost function is chosen as

$$J_{\mathrm{cpc}}(\rho) = \frac{1}{2N} \sum_{k=1}^{N} \mathrm{E}\left[ (\Omega_{\mathrm{gen}}(k,\rho) - \Omega_{\mathrm{ref}}(k))^2 + \lambda L_{\theta}\theta(k,\rho)^2 \right], \qquad (36)$$

where the input weighting factor $\lambda$ and the step size $\gamma$ in the parameter update rule (2) are both considered for different values. Note that the input signal in the cost function is high-pass filtered by $L_{\theta}$, which is identical to (32). The high-pass filter $L_{\theta}$ is required because the optimization procedure should focus on the dynamic pitch response rather than on the static pitch offset required to maintain the rated generator speed.

## 5.2  Results of IFT for CPC

The first result is obtained by optimizing the CPC for a turbulent wind field with mean wind speed $14\,\mathrm{m\,s}^{-1}$ and turbulence intensity of $4\,\%$. The step change for this result is chosen to be $\Omega_{\mathrm{step}} = 30\,\mathrm{rpm}$. Moreover, the adjustable signal-to-noise ratio gain $F$ is set to 0.02. The initial PI controller values are $K_p = 4 \cdot 10^{-3}$ and $K_i = 1 \cdot 10^{-3}$. The results for three cases with varying input weighting $\lambda$ and step size $\gamma$ are shown in Figure 14.

The trajectory of the cost function values and the controller parameters are shown in Figure 14. It can be observed that the initial controller parameters were suboptimal and converge in a few iterations to a much better performance. The results also show that the convergence of the parameters behave differently. The proportional gain $K_p$ has converged to its final value after four iterations. The integral controller gain $K_i$, on the other hand, slowly increases to larger values. The difference in trajectory can be explained due to the fact that in the step response the proportional gain is more dominant. In order to make the integral
controller parameter more dominant, one could increase the experiment length $N$.

The effect of the gains $\gamma$ and $\lambda$ on the convergence trajectories are apparent in Figure 14. It is observed that the convergence of the proportional gain $K_p$ for the cases where $\lambda = 5 \cdot 10^3$ is faster due to the higher step size $\gamma$. The effect of increasing



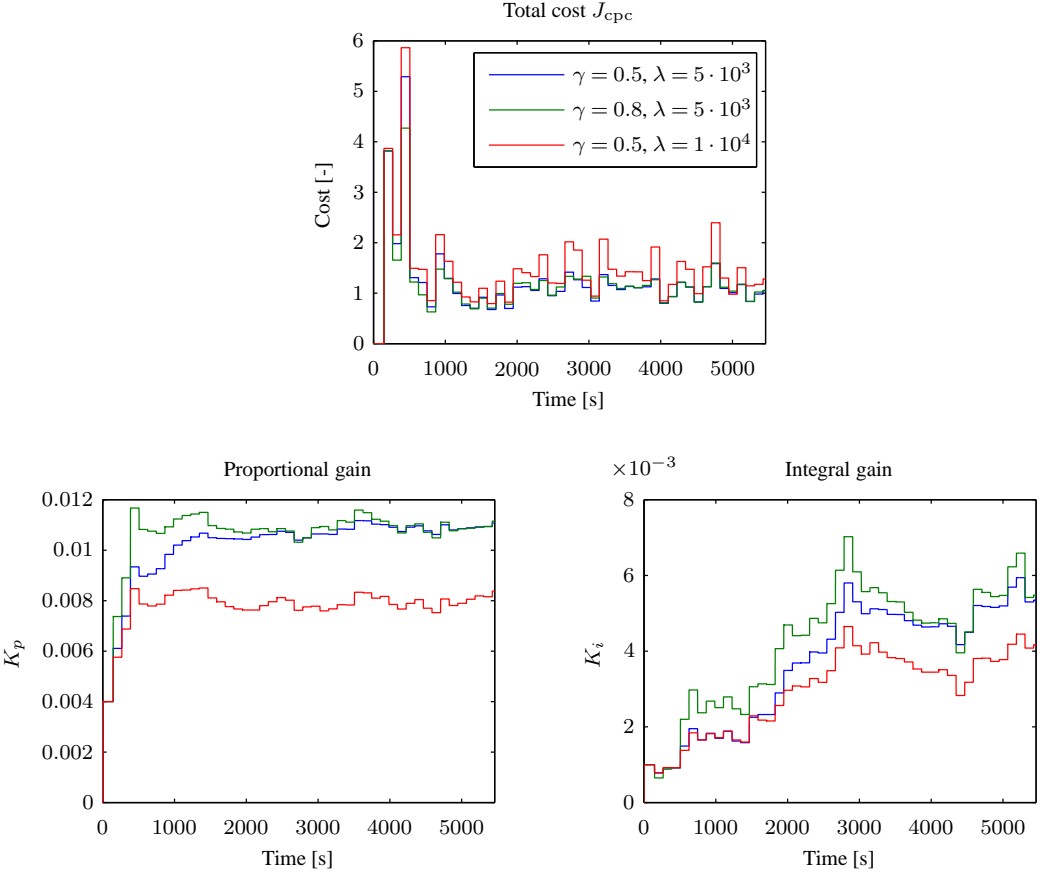

**Figure 14.** Reference step tuning using IFT of CPC at a wind speed of $14\,\mathrm{m\,s^{-1}}$ with $4\,\%$ turbulence intensity.. The results show three different cases where the step size $\gamma$ and the input weighting $\lambda$ in the cost function are varied.

the weight $\lambda$ in the cost function is also clear from Figure 14: the parameters converge to smaller values, which is according expectation.

In Figure 15, three different step responses are shown. The blue graph displays the response of the initial controller to the step change. As can be observed, this response is sloppy and after nine iterations has improved to a decent response. The increased weight on the input cost yields a step response where less pitch duty is required with only a very limited loss of tracking performance. In Figure 16, the generator speed response and the collective pitch angle for a full iteration during the first and ninth iteration are displayed.

The final results involve a comparison between IFT of CPC for different two different wind speeds: $14\,\mathrm{m\,s^{-1}}$ and $18\,\mathrm{m\,s^{-1}}$ with $4\,\%$ turbulence intensity. It is generally known that the control authority of CPC increases when the blades are further pitched from the wind. This means that less pitch effort is required to keep the rotor speed close to rated. The IFT tuning results also display this behavior. In Figure 17 it can be seen that the proportional gain for $18\,\mathrm{m\,s^{-1}}$ is roughly two thirds the value





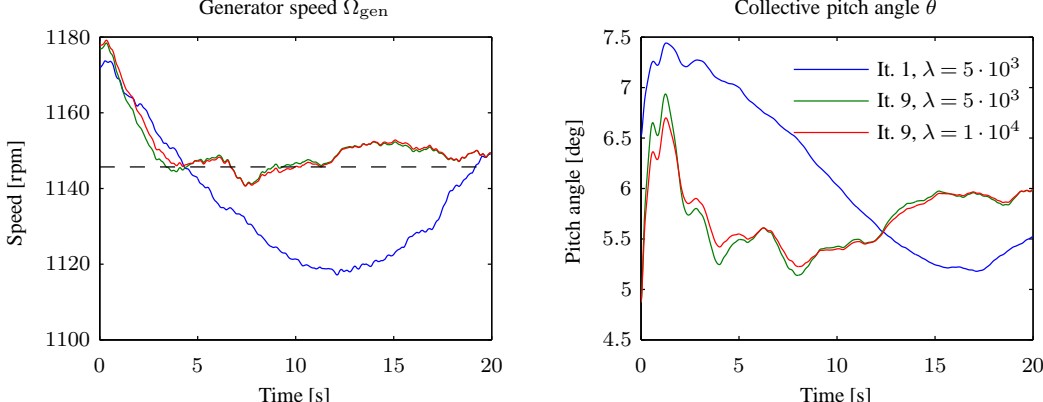

**Figure 15.** Comparison of the generator speed step response during first and ninth iteration, and for different input weighting factors $\lambda$. Results shown are obtained for a wind speed of $14\,\mathrm{m\,s^{-1}}$ with $4\,\%$ turbulence intensity.

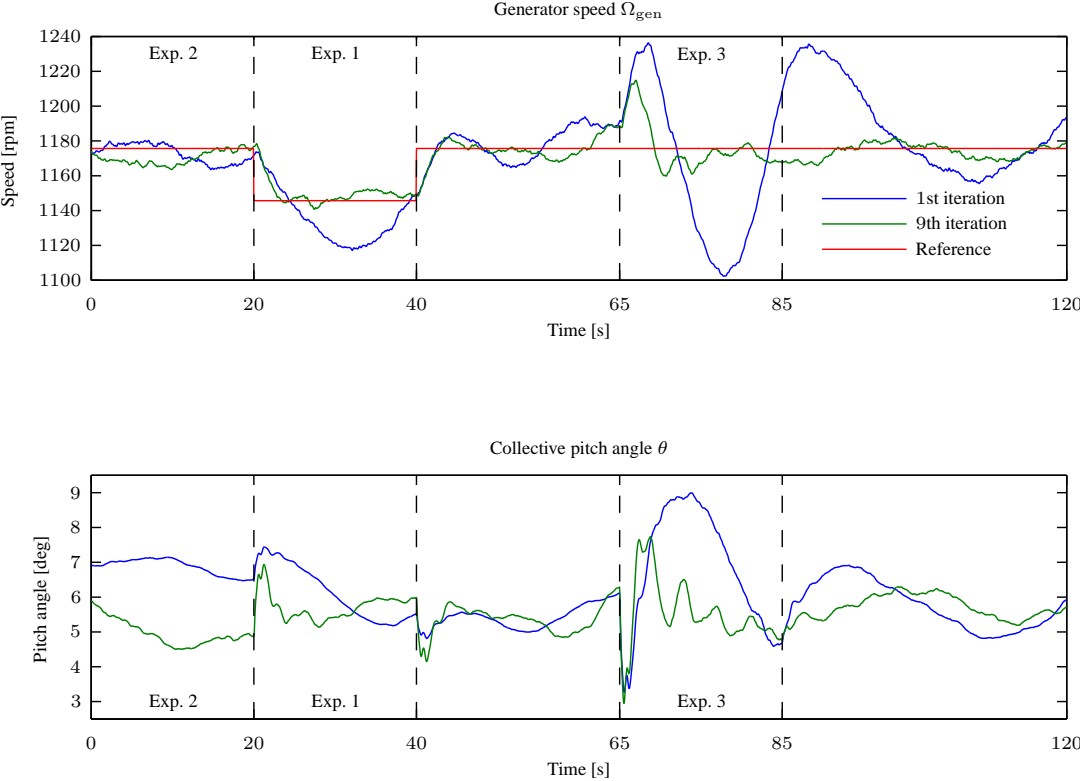

**Figure 16.** Comparison of responses during first and ninth iteration. Results shown are obtained for a wind speed of $14\,\mathrm{m\,s^{-1}}$ with $4\,\%$ turbulence intensity. The measured generator speed and reference input are shown in the upper plot, the lower plot displays the pitch response.



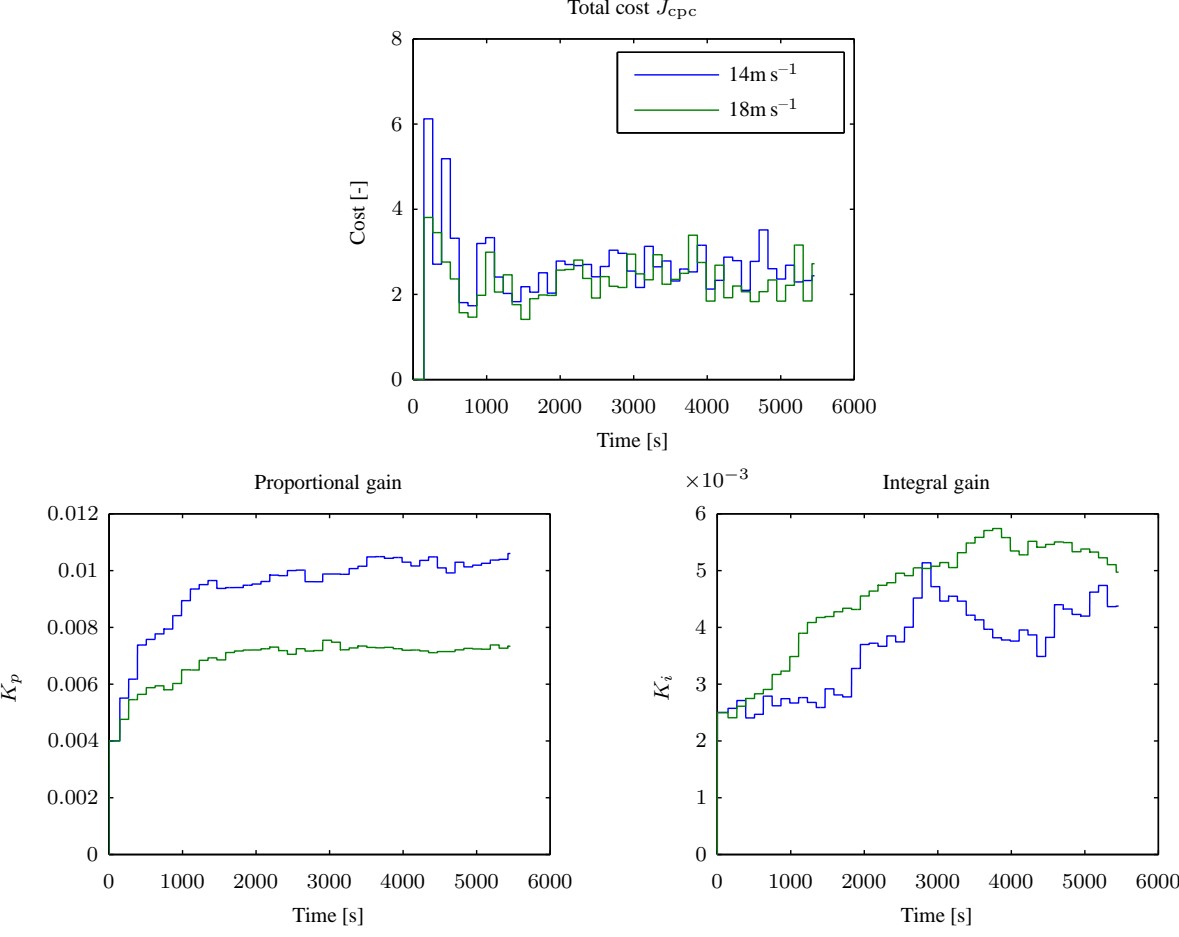

**Figure 17.** Reference step tuning of CPC using IFT of CPC at a wind speed of $14\,\mathrm{m\,s^{-1}}$ with $4\,\%$ turbulence intensity..

compared to the $14\,\mathrm{m\,s^{-1}}$ case. On the other hand, the integrator gain $K_i$ is somewhat higher for the $18\,\mathrm{m\,s^{-1}}$ wind speed. The cost function converges to a comparable result. The step responses shown for the seventh iteration are also rather similar.

## 6 Conclusions

In this paper, IFT controllers for wind turbines have been developed. The typical controller configurations used for wind turbine
5 control require three closed-loop experiments to be carried out. With the data that is collected during these experiments it has been shown that IFT can be successfully applied. The results indicate that starting the optimization from a baseline controller with decent performance, can improve the performance already within a few iterations. It has also been shown that IFT can be applied to both disturbance rejection and reference tracking control for wind turbines. This is demonstrated by means of optimizing the drivetrain damping controller and the CPC. The methodology could similarly be applied to improve fore-




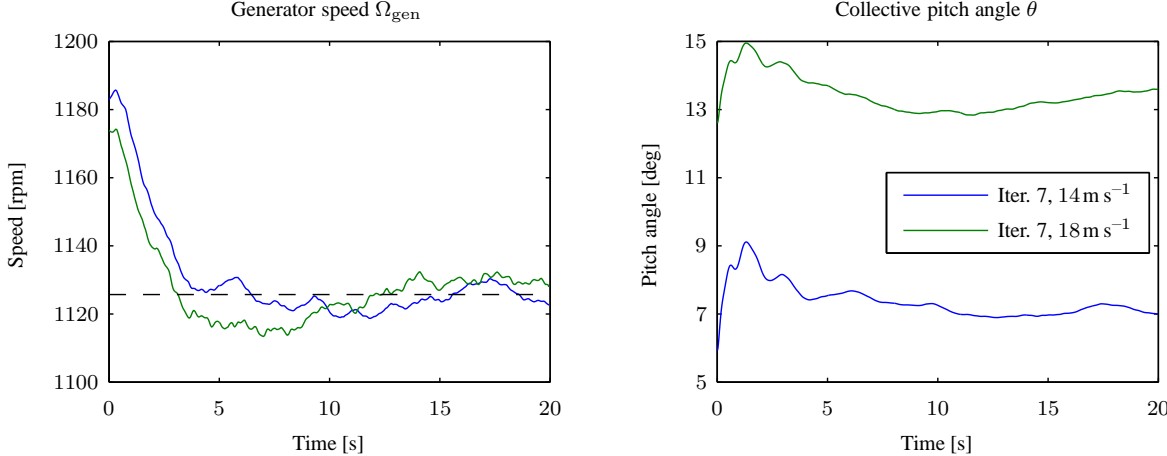

**Figure 18.** Comparison of the CPC step responses for $14\,\mathrm{m\,s^{-1}}$ and $18\,\mathrm{m\,s^{-1}}$ with $4\,\%$ turbulence intensity during the seventh iteration.

aft and/or side-side tower damping performance. Finally, it is argued that IFT could provide a valuable tool with which the performance of wind turbine controllers can be improved without the need of system identification.

## Appendix A: Derivation of input gradients

### A1 Derivation of input gradient for Section 2.3

5 Before the derivation of the input gradient is given, first recall that the control system in Section 2.3 is according to Figure 1 and therefore the sensitivity function $S(\rho)$ and complementary sensitivity function $T(\rho)$ are given by

$$S(\rho) = (I + PC(\rho))^{-1}, \tag{A1}$$
$$T(\rho) = (I + PC(\rho))^{-1}PC(\rho). \tag{A2}$$

Then, note that in Figure 1 the input $u(k, \rho)$ is determined by

10 $$u(k, \rho) = S(\rho)C(\rho)r_o + S(\rho)C(\rho)v(k), \tag{A3}$$

where $r(k)$ has been replaced by $r_o$. The gradient of (A3) with respect to $\rho$ is derived as

$$\frac{\partial u}{\partial \rho}(k, \rho) = \left( \frac{\partial S}{\partial \rho}C(\rho) + S(\rho)\frac{\partial C}{\partial \rho} \right) r_o - \left( \frac{\partial S}{\partial \rho}C(\rho) + S(\rho)\frac{\partial C}{\partial \rho} \right) v(k). \tag{A4}$$

The derivative of the sensitivity function $S(\rho)$ equals

$$\frac{\partial S}{\partial \rho} = -(I + PC(\rho))^{-2}P\frac{\partial C}{\partial \rho}. \tag{A5}$$



Substituting for the latter sensitivity derivative in (A4) yields

$$\frac{\partial u}{\partial \rho}(k,\rho) = S(\rho)\frac{\partial C}{\partial \rho}\big(r_o - T(\rho)r_o + T(\rho)v(k) - v(k)\big). \tag{A6}$$

Realizing that $(T(\rho) - I)v(k) = -S(\rho)v(k)$ and that $T(\rho)r_o + S(\rho)v(k) = y(k,\rho)$ gives the input gradient

$$\frac{\partial u}{\partial \rho}(k,\rho) = S(\rho)\frac{\partial C}{\partial \rho}(r_o - y(k,\rho)). \tag{A7}$$

## 5  A2  Derivation of input gradient for Section 2.4

The derivation of the input gradient in Section 2.4 is similar to the derivation in Appendix A.1. Here, the sensitivity $S(\rho)$ and complementary sensitivity $T(\rho)$ are as follows

$$S(\rho) = (I + PC_1 - PC_2(\rho)\mathcal{H})^{-1}, \tag{A8}$$

$$T(\rho) = (I + PC_1 - PC_2(\rho)\mathcal{H})^{-1}PC_1. \tag{A9}$$

10  In Figure 4, the input $u(k,\rho)$ equals

$$u(k,\rho) = S(\rho)C_1 r(k) + S(\rho)C_2(\rho)\mathcal{H}v(k) - S(\rho)C_1 v(k), \tag{A10}$$

such that the gradient can be derived to be

$$\frac{\partial u}{\partial \rho}(k,\rho) = \frac{\partial S}{\partial \rho}C_1 r(k) + \left(\frac{\partial S}{\partial \rho}C_2(\rho)\mathcal{H} + S(\rho)\frac{\partial C_2}{\partial \rho}\mathcal{H} - \frac{\partial S}{\partial \rho}C_1\right)v(k), \tag{A11}$$

where

15  $$\frac{\partial S}{\partial \rho} = (I + PC_1 - PC_2(\rho)\mathcal{H})^{-2}P\mathcal{H}\frac{\partial C_2}{\partial \rho}. \tag{A12}$$

With the derivative of the sensitivity function, the input gradient becomes

$$\frac{\partial u}{\partial \rho}(k,\rho) = S(\rho)\frac{\partial C_2}{\partial \rho}\mathcal{H}\Big(S(\rho)PC_1 r(k) + \big(S(\rho)PC_2(\rho)\mathcal{H} - S(\rho)PC_1 + I\big)v(k)\Big). \tag{A13}$$

Realizing that $\big(S(\rho)PC_2(\rho)\mathcal{H} - S(\rho)PC_1 + I\big)v(k) = S(\rho)v(k)$, yields the final result

$$\frac{\partial u}{\partial \rho}(k,\rho) = S(\rho)\frac{\partial C_2}{\partial \rho}\mathcal{H}\Big(T(\rho)r(k) + S(\rho)v(k)\Big) = S(\rho)\frac{\partial C_2}{\partial \rho}\mathcal{H}y(k,\rho) = S(\rho)\frac{\partial C_2}{\partial \rho}y'(k,\rho). \tag{A14}$$



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
