# Peer review of "Iterative feedback tuning of wind turbine controllers"

_Wind Energy Science, 2016_

## Referee Comment (RC1) · T. Knudsen (Referee) · 18 Jun 2016

**Review - Iterative feedback tuning of wind turbine controllers**

Torben Knudsen

June 18, 2016

**Summary**

This paper gives a good presentation of an interesting alternative way of tuning wind turbine controllers.

- The contributions are not stated perfectly clearly. "The main contribution is therefore to show the (practical) application of IFT.." is clear but regarding the vaguely formulated contributions to IFT it is hard to understand if these are new results or already is in existing literature.

- More precise statements on the practical usefulness of the methods and cases demonstrated would be nice.

- There are also a number of minor issues that needs clarification.

Consequently a revision is recommended before publication.

**Specific comments**

- Abstract and Introduction section are good.

- P2 "The main contribution is therefore to show the (practical) application of IFT to existing wind turbines." Is this not a contribution of Navalkar and van Wingerden, 2015? Moreover, are the contributions in this paper non overlapping with the ones in Navalkar and van Wingerden, 2015?

- Eq (1) If $r$ is constant the system is stationary so the expectation in (1) would be time independent. Why is the time averaging sum included? Is possible $r$ time dependence assumed known? What would be the case for this application?

- P3 "It is clear that minimizing (1) boils down to computing the gradient $\partial J(\rho_i)/\partial\rho$ and Hessian $R_i$ at every iteration." This assumes a convex problem? How do you know assures this?

- Eq (3)

  - Why doesn't $\partial J(\rho)/\partial\rho$ depend on $r$ ?
  - The expectation operator E is not well defined as long as the stochastic part $v$ of the model is not well defined.

- Eq (3)-(10) Explain how you handle the noise and expectation? Maybe you derive the gradients assuming no noise i.e. a deterministic system and then uses them even though there are noise. If so, is this correct?

- P6 2.4 IFT for systems with multiple controllers. You are referring to the drive train damping controller (DTDC) being active below rated where the generator torque is controlling the speed. Below rated the DTDC is normally not really needed because the speed controller ads damping in contrast to above rated where the generator torque/power is constant which gives no/negative damping.

- P10 "The natural frequency of the drive train dynamics... is at $\omega_r = 10.49rads^{-1}$" What is the corresponding damping which is equally important?

- P10 Why are you using two inputs $\theta_1, \theta_2$ for collective pitch control?

- P10 Why do you choose Bladed over FAST?

- P11 "The controllers are discretized using the Tustin approximation and run at a sampling time of 0.01 s." Why using 100Hz sampling frequency when the DTD dynamics is having a frequency around 1.67Hz?

- P12 "in Bode diagram, it is recognized that the band pass gain K can be increased infinitely" In practice the torque "actuator" will have a high bandwidth and a (communication delay) in the order of milliseconds seconds so there will be a limit to the gain.

- P15 4.4.1 General analysis of results. "The wind field considered here has a mean wind speed of 14 m/s" Then this is mostly above rated (11.3 m/s) where the pitch controller is active and generator power is at rated. Is this the intention?

- P16 "4.4.3 Varying experiment length $N$" I am still confused about $N$. You can not really calculate the expectation in (1). Does the $N$ amount to time averaging instead of ensemble averaging i.e. assuming ergodicity which might be OK?

- P22 "outputs a demanded collective pitch signal $\theta = \theta_1 = \theta_2$" Now suddenly I guess you assume a two bladed turbine? I assume the FAST 5MW turbine is three bladed as you also state in table 1?

- P22 "The IFT algorithm is applied so as to optimize the step response tracking of the controller in (35)." What is the relevance of a step change in speed in practice? I would think a step in power reference is more useful for derating wind farms or for primary frequency control.

- P23 5.2 Results of IFT for CPC. It is clear that the IFT method works as expected. However, the improvement over more traditional and simple tuning is not so clear as it is not easy to see what is done to do a fair simple tuning of the controller. For example regarding changing average wind speed a traditional CPC controller would also include some gain scheduling to change decrease the gain with increasing wind speed.

- P23 5.2 Results of IFT for CPC. In general it is problematic to compared two control design methods which can be tuned for different purposes/objectives and claim that one is superior. One way to do a more fair comparison is to use Pareto curves see e.g. Odgaard et al. [2015b,a].

**References**

P. F. Odgaard, T. Knudsen, and R. W. T. Bak. Optimized control strategy for over loaded offshore wind turbines. In *Proceedings of EWEA Offshore 2015*. EWEA, EWEA?, 2015a.

P. F. Odgaard, T. Knudsen, A. Overgaard, H. Steffensen, and H. Jørgensen. Importance of dynamic inflow model predictive control of wind turbines. In *9th IFAC Symposium on Control of Power and Energy Systems CPES 2015, New Delhi, India, December 9-11 2015*, volume 48 of *IFAC Workshop Series*, pages 90–95. IFAC, IFAC, 2015b. doi: 10.1016/j.ifacol.2015.12.359.

---

## Referee Comment (RC2) · M. Mirzaei (Referee) · 26 Jul 2016

**A review on "Iterative feedback tuning of wind turbine controllers"**

Mahmood Mirzaei, DTU-Wind Energy

July 26, 2016

**Comments**

The paper addresses an interesting topic on tuning (or fine-tuning) of wind turbines in operation. As it is correctly mentioned in the paper, three factors (or probably more) affect the performance of a controller designed for a simulation model, when implemented on an actual wind turbine. **1)** Model discrepancies due to modeling errors and/or manufacturing errors, **2)** The characteristics (in this case dynamics) of a wind turbine changes over time and therefore the controller needs to adjust to the new dynamics.

1) In this paper, the IFT is not used to tune a controller and track changes of the wind turbine, rather it is used to tune a controller and compare it against a base-line controller. I think it adds to the value of the paper to also support the second claim. It'd be interesting to induce some changes in the dynamics of the wind turbine and let the IFT re-tune the controller to the new changes. This can be for example changes in the aerodynamics of the blades due to e.g. leading edge erosion etc.

2) I might have missed it, but how do the authors support the first claim that the IFT method is re-tuning the controller for model discrepancies. In fact the controllers are implemented on the same simulation model. It would be interesting to test the algorithm on a simulation model whose characteristics are different from the design model (introducing some modeling/ manufacturing errors into account) and then let the IFT re-tune the controller for a better performance.

3) Is it relevant to check stability of the system when IFT is used for tuning? The authors have mentioned that they have not encountered stability issues, but does it guarantee that the system will stay stable?

4) The controller for the drive-train damper does not show significantly improved results over the base-line controller and the implementation of IFT on the pitch controller improves performance for set point tracking.

5) The band-pass gain in figure 9 does not seem to converge during the experiment. The same goes to some other parameters that are adjusted by IFT.
    a. What could be the reason for that?
    b. In general it doesn't seem that the parameters converge to a steady state value after 6000s of simulation. Shouldn't they finally converge?
    c. The parameters of the simulation model haven't changed, so shouldn't the parameters of the controller finally settle to specific values?
    d. Doesn't this raise an issue with stability of the method?

6) It is mentioned that the pitch controller is designed for step response tracking. It is known that wind turbine controllers in the full load region are designed for disturbance rejection with, rather than set-point tracking. So, the following questions arise:

   a. Is it relevant at all to design a wind turbine controller for step response tracking of the rotational speed?
   b. What's the applicability of set-point tracking for a wind turbine controller?
   c. What about the performance of the controller for disturbance rejection? What is the effect of re-tuning the controller for set-point tracking on disturbance rejection performance?
   d. Even if we are to improve performance of the wind turbine controller for set-point tracking, wouldn't it be easier to use a 2-DOF controller instead, so that we are not sacrificing the important disturbance rejection performance?
   e. Following the point mentioned above, it'd be better to compare a controller designed for set point tracking (design using IFT), against the same type of controller (e.g. a tuned 2-DOF wind turbine controller).

7) The same issue with convergence of the controller parameters exists for the pitch controller. It seems that the $K_i$ of the PI controller is not converging to a specific value after over 5000s simulation time.

**Minor comments**

1) It'd be better to explain more clearly in the abstract that this algorithm is in fact an online tuning method.
2) In page 14, line 12 the authors mention "an inverted notch filter." Does this mean a band-pass filter?

---

## Author Comment (AC2) · 9 Feb 2017

Dear Editor/Reviewers,

We as authors would like to thank the editor and reviewers for their valuable contribution in improving the manuscript. The last comment is indeed correct and we performed a complete revision of the paper to make sure that the issue is now corrected. You can find the final version of the abstract and manuscript in the attachment.

One small question remains: can you please update the authors of this submission according to the authors stated in the heading of the manuscript?

Regards, Edwin van Solingen Sebastiaan Paul Mulders Jan-Willem van Wingerden

[Figure]

Please also note the supplement to this comment:
http://www.wind-energ-sci-discuss.net/wes-2016-7/wes-2016-7-AC2-supplement.zip

---

## Author Response (AR1)

| Date | December 2, 2016 |
| Our reference | n/a |
| Your reference | n/a |
| Contact person | S.P. Mulders |
| Telephone/fax | +31 (0)6 5573 6149 / n/a |
| E-mail | S.P.Mulders@TUDelft.nl |
| Subject | Response to reviewers |

[Figure]

**Delft University of Technology**

Delft Center for Systems and Control

Address
Mekelweg 2 (3ME building)
2628 CD Delft
The Netherlands

www.dcsc.tudelft.nl

Reviewers
*Wind Energy Science*

Dear Reviewers,

First of all, the authors would like to thank the reviewers for their positive and constructive feedback. We believe that the comments will help us to significantly improve the quality of the paper. The objective of this document is to respond to the points raised by the reviewers (blue) and to provide an overview of the changes that are made to the paper (red). The document consists of two sections, each addressing the comments of the reviewers separately.

Yours sincerely,

Edwin van Solingen
Sebastiaan Paul Mulders
Jan-Willem van Wingerden

Enclosure(s):  Response to comments of Reviewer 1
Response to comments of Reviewer 2

**Response to comments of Torben Knudsen**

**Reviewer 1 comments**: This paper gives a good presentation of an interesting alternative way of tuning wind turbine controllers.

1. The contributions are not stated perfectly clearly. The main contribution is therefore to show the (practical) application of IFT.. is clear but regarding the vaguely formulated contributions to IFT it is hard to understand if these are new results or already is in existing literature.
   Thanks for raising this remark. It is important to note here that the IFT method in general has been well studied. However, we contribute in the sense of applying IFT in presence of other controllers and dealing with systems operating at offsets.

   In the revised document this is made more clear in Section 1: "This paper ... convergence speed.".

2. More precise statements on the practical usefulness of the methods and cases demonstrated would be nice.
   The usefulness of the methods can for example be found in fine-tuning controllers of wind turbines. This can for example be done for the example cases in the manuscript. On the other hand, another well-known application case could be the optimization of the tower damping controllers (as tower frequencies in practice often differ from the designed ones).
   The example cases in this manuscript reflect cases with turbulent wind and can, in that sense, be regarded as practical. Of course, with the IFT approach in our manuscript, the closed-loop system could become unstable after each controller parameter update. Thus before applying on a real turbine, safety considerations should be taken into account. The presented algorithm itself can be directly applied without modification to a real turbine.

   Remarks with respect to practical usefulness are extended and better explained and/or motivated in Section 1: "Another strategy... turbine lifetime).", "The key... not needed.". Section 2.6: "However, a practical... the cost function gradient.". Conclusion: "The methodology... system identification."

3. P2 "The main contribution is therefore to show the (practical) application of IFT to existing wind turbines." Is this not a contribution of Navalkar and van Wingerden, 2015? Moreover, are the contributions in this paper non overlapping with the ones in Navalkar and van Wingerden, 2015?

In the paper of [1], LPV IFT in a feedforward setting is studied, which is not yet very practical. On the contrary, in this paper we look at the tuning of existing feedback controllers using linear IFT.

To address the comment, some words about the differences of both papers are added in the edited manuscript. Section 1: "In the past decade... roughly unaffected"

4. Eq. (1) If $r$ is constant the system is stationary so the expectation in Eq. (1) would be time independent. Why is the time averaging sum included? Is possible $r$ time dependence assumed known? What would be the case for this application?
   The basic rationale of IFT is to minimize a cost function as in Eq. (1). In literature, this is typically done by optimizing/fitting the systems step response with a predefined and desired step response. Moreover, the system will only be time independent in case there is no disturbance acting on the system (or can be assumed negligible) with the reference being constant. Typically, however, the system is operated in the presence of noise, which is the reason of including the expectation. The cases for this application can for example be the example cases shown in Section 5.

   An explanation is included in section 2.1: "and involves... to noise."

5. P3 It is clear that minimizing (1) boils down to computing the gradient $\partial J(\rho_i)/\partial \rho$ and Hessian $R_i$ at every iteration. This assumes a convex problem? How do you know assures this?
   The underlying problem is a non-convex optimization problem, and with the proposed method the gradient search converges to a local minimum. Despite this, there is numerous of literature available presenting successful results using this optimization approach (a number of studies are also referenced in the manuscript).

   In the revised manuscript this is indicated in Section 2.1: "Please note... local minimum."

6. Eq. (3) Why doesn't $\partial J(\rho_i)/\partial \rho$ depend on $r$? The expectation operator $E$ is not well defined as long as the stochastic part $v$ of the model is not well defined
   The reason why the derivative of the cost function does not depend on the reference input is because the reference input does not depend on the controller parameters. Hence, taking the derivative of the reference input to the controller parameters is zero.
   The reviewer is right in stating that the noise is not yet defined at that point in the manuscript. However, the expectation operator is generally defined and its definition does not change with different noise types. Assumptions on the noise are given at a later stage in the manuscript.

   The manuscript is extended in Section 2.1: "where $P$ is... measurement noise.".

7. Eq. (3)-(10) Explain how you handle the noise and expectation? Maybe you derive the gradients assuming no noise i.e. a deterministic system and then uses them even though there are noise. If so, is this correct?

Thank you for addressing this topic. The derivation is done according to Eq. (3)-(10). First, when taking the derivative with respect to the controller parameters, all noise terms disappear. Then, when obtaining the derived derivative by the closed-loop experiments, this can of course not be done without noise contamination of the signals. Thus, the required (measured) estimates from the closed-loop experiments do contain noise. However, by then assuming some mild assumptions as explained in the manuscript, and taking the expected value, it can be shown that the estimates are unbiased. Concluding, the signals measured and used in each iteration do contain noise, but under the given assumptions yield unbiased estimates, due to terms that nullify as the expectation is calculated. We will not include this last step where the expectation operator is applied, as it will lengthen the manuscript and is rather straightforward, and can be found in the mentioned citations as well.

See Section 2.1: "and involves... to noise."

8. P6 2.4 IFT for systems with multiple controllers. You are referring to the drivetrain damping controller (DTDC) being active below rated where the generator torque is controlling the speed. Below rated the DTDC is normally not really needed because the speed controller ads damping in contrast to above rated where the generator torque/power is constant which gives no/negative damping.

Indeed, the authors are referring to the below-rated scenario where both DTDC and power control are active. We believe that it is strongly turbine dependent whether both controllers will/need to be active in below-rated conditions. However, regardless whether DTDC is used in below-rated or not, the considered scenario could be applied similarly to other loops, e.g., the fore-aft damping control and collective pitch control.

colorred In the revision we will make a remark about the turbine dependence with respect to DTDC and power control being active at the same time and that the method can be applied to other scenarios (such as fore-aft damping and CPC) as well. Therefore, see Section 4.4.4: "Although the need... is active.". Conclusion: "The methodology... system identification.".

9. P10 "The natural frequency of the drive train dynamics. . . is at $\omega_r = 10.49\,\text{rad/s}$." What is the corresponding damping which is equally important?

The authors agree that this is important information that was missing. The damping ratio of the drivetrain mode in Fig. 7 is $0.0217$.

The damping value is mentioned in the revision in Section 3: "The natural frequency... , respectively.".

10. P10 Why are you using two inputs $\theta_1, \theta_2$ for collective pitch control?
Thanks for pointing out this mistake. This should be just $\theta$.

The mistake is corrected in Figure 6

11. P10 Why do you choose Bladed over FAST?
Bladed and FAST are both widely used and acknowledged wind turbine software packages. The choice for either of them is in that sense rather arbitrary: the achieved results can be reconstructed using either package. We have chosen for Bladed for a practical reason: we had the NREL 5MW model and a baseline controller available in Bladed.

12. P11 "The controllers are discretized using the Tustin approximation and run at a sampling time of $0.0\,s$." Why using 100Hz sampling frequency when the DTD dynamics is having a frequency around 1.67Hz?
As the reviewer correctly points out, the controller frequency is much higher than required. Again, the results could have equally well have been achieved with much lower sampling time. However, as the algorithm was implemented on an existing baseline controller, the sampling time was kept to 0.01s.

In the revised manuscript a statement on this is made in Section 3: "Note that... this frequency.".

13. P12 "in Bode diagram, it is recognized that the band pass gain K can be increased infinitely In practice the torque actuator will have a high bandwidth and a (communication delay) in the order of milliseconds seconds so there will be a limit to the gain.
The reviewer is right. The observation drawn in the original manuscript indeed assumes perfect actuation.

We have adjusted this in our revision, Section 3: "From the open-loop... unmodeled dynamics.".

14. P15 4.4.1 General analysis of results. "The wind field considered here has a mean wind speed of 14 m/s." Then this is mostly above rated (11.3 m/s) where the pitch controller is active and generator power is at rated. Is this the intention?
This is indeed intentional. For the considered wind speed the drivetrain is easily excited and makes it therefore an interesting example case. Again, also lower or higher wind speeds could be considered, which yield a similar outcome.

15. P16 "4.4.3 Varying experiment length $N$." I am still confused about $N$. You can not really calculate the expectation in (1). Does the $N$ amount to time averaging instead of ensemble averaging i.e. assuming ergodicity which might be OK?

Here we work with time-averaging and have to deal with the typical trade-off in system identification where the variance is inversely proportional with the amount of data used (in this case to estimate the gradient).

16. P22 "outputs a demanded collective pitch signal $\theta = \theta_1 = \theta_2$." Now suddenly I guess you assume a two bladed turbine? I assume the FAST 5MW turbine is three bladed as you also state in table 1?

Thanks once more for pointing out a similar mistake as before. To be very clear at this point: we have considered the *three-bladed* NREL 5MW.

This is fixed in the new version, see Sections 3 and 5.1.

17. P22 "The IFT algorithm is applied so as to optimize the step response tracking of the controller in (35)." What is the relevance of a step change in speed in practice? I would think a step in power reference is more useful for derating wind farms or for primary frequency control.

We agree with the reviewer that only tuning the step response to a generator speed change might not be very relevant. However, as the reviewer indicates, indeed in derating wind turbines (or farms), a step might occur. But, more importantly, the presented step response approach not only optimizes the step response, but also implicitly optimizes disturbance rejection. The goal of this section has mainly been to show the flexibility of the framework.

In the revised version a further elaboration of the relevance is included, by putting the step response approach into perspective and discussing the implicit inclusion of disturbance rejection. See Section 5.1: "As CPC... is uncontrollable."

18. P23 5.2 Results of IFT for CPC. It is clear that the IFT method works as expected. However, the improvement over more traditional and simple tuning is not so clear as it is not easy to see what is done to do a fair simple tuning of the controller. For example regarding changing average wind speed a traditional CPC controller would also include some gain scheduling to change decrease the gain with increasing wind speed.

We believe that the results should be put into a broader perspective. Having a fairly simple tuning method at hand that can 'automatically' fine-tune controllers due to e.g., changing wind turbine dynamics, or 'finding' the tower eigenfrequency (as tower frequencies in practice often differ from the designed ones), can have an impact. Besides, the method is not intended as to replace other or existing methods, but rather as an addition that further tunes the controller online.

To do a simple tuning of the controller resulting performance that is satisfactory on the actual system, a (linear) model obtained from first principles modeling or system identification is needed. With IFT, one might assume a simple model, which at least stabilizes the system. Automatic and online tuning of the controller, resulting in a performance level expressed in an objective function, can be performed on the actual turbine.

Moreover, with respect to the gain scheduling, we would like to refer to the paper of [1]. In this study, using similar techniques as described in our manuscript, the optimal gain scheduling is learned.

19. P23 5.2 Results of IFT for CPC. In general it is problematic to compared two control design methods which can be tuned for different purposes/objectives and claim that one is superior. One way to do a more fair comparison is to use Pareto curves see e.g. Odgaard et al. [2015b,a].
We acknowledge this remark. Indeed, Pareto curves can be used to do a comparison between the different controllers. However, the purpose of this paper is not to tune controllers to obtain the most optimal controller (this would be a large optimization problem taking into account all other controllers on the one side and loads on the other side), but rather to demonstrate that IFT can be used to re-tune, or fine-tune/further optimize controllers from an initial controller design, to meet a desired performance objective expressed in a cost function.

In the revision we will discuss the possibility of using Pareto curves and cite the work of Odgaard et al. [2015b,a]. We will also clarify any remaining confusion concerning the comparison of the controller performances. See Section 5.2: "The trajectory... will not be considered.".

**Response to comments of Mahmood Mirzaei**

**Reviewer 2 comments**: The paper addresses an interesting topic on tuning (or finetuning) of wind turbines in operation. As it is correctly mentioned in the paper, three factors (or probably more) affect the performance of a controller designed for a simulation model, when implemented on an actual wind turbine. 1) Model discrepancies due to modeling errors and/or manufacturing errors, 2) The characteristics (in this case dynamics) of a wind turbine changes over time and therefore the controller needs to adjust to the new dynamics.

- In this paper, the IFT is not used to tune a controller and track changes of the wind turbine, rather it is used to tune a controller and compare it against a baseline controller. I think it adds to the value of the paper to also support the second claim. It'd be interesting to induce some changes in the dynamics of the wind turbine and let the IFT retune the controller to the new changes. This can be for example changes in the aerodynamics of the blades due to e.g. leading edge erosion etc.
  The authors thank the reviewer for the valuable comment. We believe that the method is inheritably capable of doing so. This can for example be recognized from Eq. (3) in the manuscript. It is clear that the objective function depends on the dynamics of the wind turbine, and, therefore when the turbine dynamics change, due to e.g. leading edge corrosion, consequently also the objective function and optimization change. We will incorporate changing dynamics in future research.

- I might have missed it, but how do the authors support the first claim that the IFT method is re-tuning the controller for model discrepancies. In fact the controllers are implemented on the same simulation model. It would be interesting to test the algorithm on a simulation model whose characteristics are different from the design model (introducing some modeling/manufacturing errors into account) and then let the IFT re-tune the controller for a better performance.
  This follows a similar reasoning as applies to the former remark posed by the reviewer. The re-tuning of the controller is inherent to minimizing the given objective function: when the underlying plant changes, the output and input change accordingly, and, hence, a different control tuning will be obtained.

  This is clarified in Section 3: "Initial (baseline) controllers... to a DLL file.".

- Is it relevant to check stability of the system when IFT is used for tuning? The authors have mentioned that they have not encountered stability issues, but does it guarantee that the system will stay stable?

We agree with the reviewer that this is a misleading statement in the original manuscript. Robustness and stability is always relevant when it comes to application in practice. In our study we have not encountered stability issues, and have for that reason not explicitly considered it. However, in Section 2.6, we have briefly discussed stability and robustness, and provide the readers interested in this topic several references.

In our revised manuscript we changed the statement regarding stability considerations and included some remarks for practical application. See Section 2.6: "However, a practical... the cost function gradient.".

- The controller for the drive-train damper does not show significantly improved results over the baseline controller and the implementation of IFT on the pitch controller improves performance for set point tracking.
  We partly disagree with the reviewer's remark. Indeed, the differences are relatively small compared to the baseline controller. This is mainly caused by the fact that the baseline controller is performing already quite well. For that reason the case where a poorly tuned initial controller is used as a starting point is also included. For this case, the IFT algorithm converges close to the baseline controller, supporting our claim that the baseline controller is rather 'optimal'.

  In the revision this is included in Section 4.4.1: "From the three plots... roughly 10 iterations.".

- The band-pass gain in figure 9 does not seem to converge during the experiment. The same goes to some other parameters that are adjusted by IFT.
  a. What could be the reason for that?
  b. In general it doesn't seem that the parameters converge to a steady state value after 6000s of simulation. Shouldn't they finally converge?
  c. The parameters of the simulation model haven't changed, so shouldn't the parameters of the controller finally settle to specific values?
  d. Doesn't this raise an issue with stability of the method?
  The authors thank the reviewer for raising these remarks and questions. The underlying reason for the questions and remarks is the fact that the observed results are driven by the turbulent wind. As is mentioned in the manuscript, a turbulence intensity of 10% is used. This means that for the considered iteration length $N$, the objective function and estimated gradient signals change in every iteration. For that reason, the parameters remain varying around the steady-state values. This can especially be observed when the simulation length is increased. Moreover, when the turbulence intensity is decreased, the parameter variations decrease. Ultimately, with constant wind, the parameter values converge to a fixed value.

In order to support our reasoning above, please refer to the figures at the end of this document, where the results of an extended simulation are shown. From these plots it is clear that the optimization converges and varies due to turbulence about the steady-state result.

In order to address the observations and remarks made by the reviewer, we will increase the simulation length to demonstrate that the parameters indeed have converged.

The manuscript is adjusted in Section 2.6: "However, a practical... the cost function gradient.". Section 4.4.1: "It is observed... intensity used.".

- It is mentioned that the pitch controller is designed for step response tracking. It is known that wind turbine controllers in the full load region are designed for disturbance rejection with, rather than setpoint tracking. So, the following questions arise:
  a. Is it relevant at all to design a wind turbine controller for step response tracking of the rotational speed?
  b. What's the applicability of setpoint tracking for a wind turbine controller?
  c. What about the performance of the controller for disturbance rejection? What is the effect of re-tuning the controller for set-point tracking on disturbance rejection performance?
  d. Even if we are to improve performance of the wind turbine controller for setpoint tracking, wouldn't it be easier to use a 2-DOF controller instead, so that we are not sacrificing the important disturbance rejection performance?
  e. Following the point mentioned above, it'd be better to compare a controller designed for set point tracking (design using IFT), against the same type of controller (e.g. a tuned 2-DOF wind turbine controller).
  The reviewer addresses an important aspect of collective pitch control. Indeed, typically the CPC will be designed for disturbance rejection purposes. On the other hand, often a staircase wind profile is used to check the CPC response to changes in the wind. The latter is what the authors had in mind when opting for the 'step response tuning' approach. It is important to note here that while applying a step in the generator speed setpoint, a turbulent wind field is assumed. This means that implicitly disturbance rejection also is part of the underlying optimization.

The difference between disturbance attenuation and reference tracking is the frequency spectrum of the disturbance and reference, respectively. For the application at hand, the spectrum for both scenarios are similar. Thus, a good tracking controller is also a good disturbance mitigation controller. A 2-DOF situation - we assume that the reviewer refers to the combination of feedforward and feedback - might be relevant if you have Lidar information or information of the derivatives of the reference signal. In this paper we consider a system without Lidar measurements and typically the reference signal is also uncertain over the control horizon.

We have pointed out in our revised manuscript that disturbance rejection is also implicitly included by optimization for step reference tracking. See Section 5.1: "As CPC... is uncontrollable."

- The same issue with convergence of the controller parameters exists for the pitch controller. It seems that the Ki of the PI controller is not converging to a specific value after over 5000s simulation time.
  As reasoned in one of our previous responses, due to the turbulent wind, the values vary around their steady-state values. Moreover, in this specific case, the integral gain is less dominant (compared to Kp).

  The latter is also explained in the manuscript, see Section 5.2.

**Minor comments**:

- It'd be better to explain more clearly in the abstract that this algorithm is in fact an online tuning method.
  This would indeed be better and will be done in the revised manuscript.

- In page 14, line 12 the authors mention an inverted notch filter. Does this mean a band-pass filter?
  Correct, an inverted notch filter is exactly the opposite of a notch filter and thus passing/amplifying certain frequencies.

**References**

[1] S. T. Navalkar and J. W. van Wingerden, "Iterative feedback tuning of an {LPV} feedforward controller for wind turbine load alleviation*," *IFAC-PapersOnLine*, vol. 48, no. 26, pp. 207 – 212, 2015, 1st {IFAC} Workshop on Linear Parameter Varying Systems {LPVS} 2015Grenoble, France, 7-9 October 2015.